# Sec14l3 potentiates VEGFR2 signaling to regulate zebrafish vasculogenesis

Bo Gong[1], Zhihao Li[1], Wanghua Xiao[1], Guangyuan Li[1], Shihui Ding[1], Anming Meng[1] & Shunji Jia[1]

Vascular endothelial growth factor (VEGF) regulates vasculogenesis by using its tyrosine kinase receptors. However, little is known about whether Sec14-like phosphatidylinositol transfer proteins (PTP) are involved in this process. Here, we show that zebrafish *sec14l3*, one of the family members, specifically participates in artery and vein formation via regulating angioblasts and subsequent venous progenitors' migration during vasculogenesis. Vascular defects caused by *sec14l3* depletion are partially rescued by restoration of VEGFR2 signaling at the receptor or downstream effector level. Biochemical analyses show that Sec14l3/SEC14L2 physically bind to VEGFR2 and prevent it from dephosphorylation specifically at the $Y^{1175}$ site by peri-membrane tyrosine phosphatase PTP1B, therefore potentiating VEGFR2 signaling activation. Meanwhile, Sec14l3 and SEC14L2 interact with RAB5A/4A and facilitate the formation of their GTP-bound states, which might be critical for VEGFR2 endocytic trafficking. Thus, we conclude that Sec14l3 controls vasculogenesis in zebrafish via the regulation of VEGFR2 activation.

[1] State Key Laboratory of Membrane Biology, Tsinghua-Peking Center for Life Sciences, School of Life Sciences, Tsinghua University, 100084 Beijing, China. Correspondence and requests for materials should be addressed to A.M. (email: mengam@mail.tsinghua.edu.cn) or to S.J. (email: jiasj@mail.tsinghua.edu.cn)

The vertebrate vasculature, as a tree-like tubular and highly dynamic plexus, extends into almost all tissues for a constant supply of nutrients and oxygens, or transport of metabolic wastes[1,2]. The formation of a functional vascular system is essential for embryonic development, and its structural abnormalities always lead to pathological diseases[3,4]. This systematic and hierarchical vascular system is achieved by two distinct mechanisms, vasculogenesis (de novo assembly of vessels) and angiogenesis (modification and expansion of pre-existing vessels). In zebrafish, angioblasts derived from the lateral plate mesoderm eventually give rise to the first embryonic artery (dorsal aorta, DA) and vein (posterior cardinal vein, PCV) during vasculogenesis, and then these vascular systems are rapidly expanded and remodeled during angiogenesis to consummate vessel networks, including the formation of intersegmental veins (ISV) by endothelial cell sprouting[2,5–7].

So far, a variety of signaling molecules and transcription factors have been implicated in the formation of the vertebrate vasculature via regulating endothelial cell proliferation, differentiation, migration, and position[2,4,8]. Vascular endothelial growth factor (VEGF) signaling is considered as the most critical and pivotal one during embryonic vasculogenesis as well as angiogenesis[9,10]. After secretion, VEGF ligands bind in an overlapping pattern to three receptor tyrosine kinases (RTKs), known as VEGFR1/Flt-1, VEGFR2/Flk-1/KDR, and VEGFR3/Flt-4 on the plasma membrane, followed by receptor dimerization and autophosphorylation at particular tyrosine sites. Then, the phosphorylated receptors recruit interacting proteins and further trigger the activation of downstream cascades via PLCγ/ERK and PI3K/AKT pathways[11]. Among these VEGFRs, VEGFR2 is considered as the major mediator of proangiogenic signaling in almost all aspects of vascular-endothelial-cell biology[8]. Of particular interest, in endothelial cells, VEGFR2 displays distinct distributions in subcellular pools, including cell surface, endocytic storage compartments, lipid rafts as well as cell-cell junctions[12–14]. As a result, VEGFR2 signaling could be monitored on the plasma membrane or within endosomes. However, what determines the activation of a specific pool is poorly understood[8].

To achieve specific signal outputs with coordinated duration and amplitude, VEGFR2 signaling is strictly regulated at numerous levels, such as the receptor expression level, the availability, and affinities for binding its different ligands, the presence of co-receptors and repressor (tyrosine phosphatases), and so on[15–17]. More importantly, the intracellular trafficking and endocytic kinetics of VEGFR2 also regulate the signal outputs significantly[18–20]. Upon ligand binding, VEGFR2 is internalized mainly in a clathrin-dependent manner with the help of motor proteins and trafficked to RAB5-positive and EEA1-positive early endosomes. Unless VEGFR2 is dephosphorylated by the peri-membrane resident PTP1B at the $Y^{1175}$ site[21], these VEGFR2-containing vesicles can either be targeted for degradation via the RAB7-pathway to attenuate the signaling or recycled back to the plasma membrane via the fast (RAB4-dependent) or slow (RAB11-dependent) route for further potentiation[19]. Although NRP1-synectin-myoVI and ephrinB2-DAB2-PAR3 complex have been demonstrated to promote endosome movements into cell[22,23], many events involved in VEGFR2 internalization and trafficking are still unclear. Full understanding of the endocytic VEGFR2 trafficking will further advance our understanding of the VEGF signaling and the consequential biological functions.

Sec14l3 proteins belong to phosphatidylinositol transfer proteins (PITPs), which were first described as transporters to potentiate phosphatidylinositol (PI) and phosphatidylcholine (PC) exchange between membranes in vitro[24]. Because of the involvement of PI molecules in endocytic membrane trafficking, PITPs are also proposed to play a vital role in vesicle budding

from the *trans* Golgi network (TGN) as well as in priming of exocytosis[25–27]. Although it is reported that there is a crosstalk between PITPs with Wnt/Ca$^{2+}$ and EGF signaling[28–30], whether PITPs integrate lipid metabolism with intracellular VEGF signaling has not been studied.

In this paper, we demonstrate that Sec14l3 regulates VEGF signaling at its receptor level and participates in vascular development of zebrafish embryos. Using morpholino knockdown in combination with CRISPR/Cas9 knockout system, we find that *sec14l3* deficiency results in narrowed DA and PCV lumens. Mechanistically, Sec14l3 promotes RAB5A/4A activation to accelerate VEGFR2 internalization and recycling, thus preventing VEGFR2 from exposure to the phosphatase activity of PTP1B, which ultimately enhances the downstream VEGF signaling. It will be meaningful to interpret PITPs function in regulating VEGFR2 trafficking as well as in understanding VEGF signaling during vertebrate vasculature development.

## Results

**Sec14l3/SEC14L2 are enriched in the endothelial cells.** Vascular system formation is important for the normal development of vertebrate embryos, as well as for the survival of adults[4]. In order to identify critical genes required for zebrafish vasculature development, we isolated GFP-positive cell populations, by fluorescence-activated cell sorting (FACS), from *Tg(fli1a:EGFP)$^{y1}$* embryos[7] at 24 hpf, when the DA has separated from the PCV[5,6]. RNA sequencing of the isolated cells showed that genes in vascular, hematopoietic and pharyngeal arch cells were highly expressed as reported previously (Supplementary data 1)[7]. Of our interest is the PITP gene *sec14l3*. Whole-mount in situ hybridization (WISH) results showed that *sec14l3* was spatially expressed in vasculature at 24 hpf (Fig. 1a). Transverse sectioning post WISH indicated that *sec14l3* is expressed in both arterial and venous endothelial progenitor cells at 19 hpf, and thereafter it is restricted to the PCV compartment at 25 hpf (Fig. 1a). The expression of *sec14l3* in endothelial cells is almost eliminated in *etsrp/etv2* mutants lacking trunk vasculature (Fig. 1b)[31], which confirms its vascular expression. Quantitative RT-PCR analysis indicated that *sec14l3* expression level in GFP$^+$ cells from *Tg (fli1a:EGFP)$^{y1}$* embryos at 24 hpf is enriched by about 5 folds, which is comparable to that of *kdrl/vegfr2*, a specific vascular marker gene (Fig. 1c).

To know whether the enrichment of *sec14l3* in endothelial cells is conserved in mammals, we analyzed human *SEC14L2/SEC14L3* expression in human umbilical vein endothelial cells (HUVECs). RT-PCR results showed that *SEC14L2* transcripts exist while *SEC14L3* does not (Supplementary Fig. 1a). Importantly, immunostaining data demonstrated that SEC14L2 protein locates throughout the cells, and presents as vesicles in the cytosol. Given the fact that intracellular VEGFR2 mainly distributes to endosomes and Golgi[12,32], we co-stained our protein with VEGFR2 and it showed that about 28% of SEC14L2 is co-localized with VEGFR2 in the endocytic pool in HUVECs (Fig. 1d). Taken together, these data suggest a possible implication of zebrafish *sec14l3*/human *SEC14L2* in vascular development.

**Sec14l3/SEC14L2 are required for vascular formation.** To investigate whether Sec14l3 is required for vasculature development during zebrafish embryogenesis, two antisense morpholino oligonucleotides (MO), sec14l3-tMO2 and sec14l3-sMO, were designed to knock down *sec14l3* expression in vivo. sec14l3-tMO2 targets the 5′ untranslated region of *sec14l3* mRNA to block its translation, while sec14l3-sMO targets the 2nd exon-intron junction to interrupt normal splicing of the primary transcript. Their effectiveness was confirmed by previous report[30] and

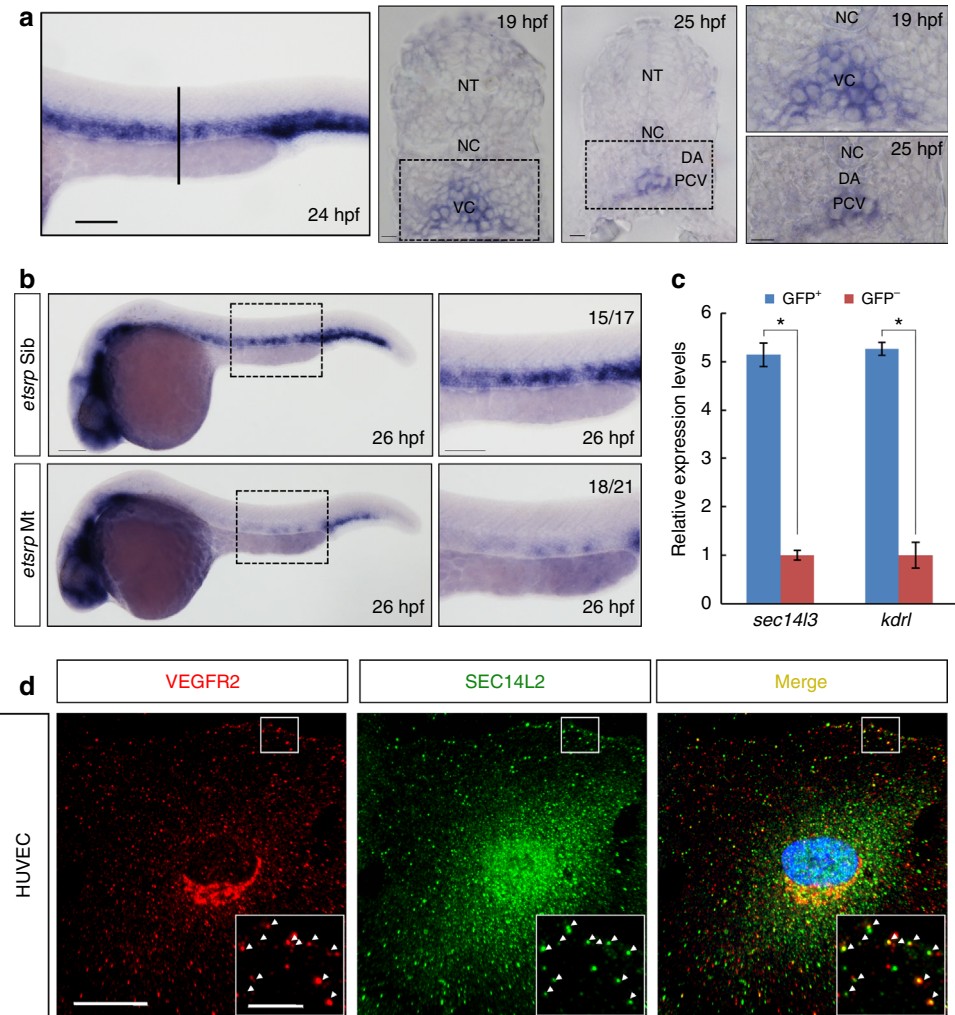

**Fig. 1** *sec14l3/SEC14L2* are expressed in endothelial cells and co-localized with VEGFR2. **a** Expression pattern of *sec14l3* examined by WISH. Trunk vascular expression of *sec14l3* is shown at 24 hpf and the vertical line denotes the equivalent position of transverse sections in embryos at 19 hpf and 25 hpf. High-resolution views of the boxed regions are shown at right panels. Adjacent tissues are denoted. NT, neural tube; NC, notochord; DA, dorsal aorta; PCV, posterior cardinal vein; VC, vascular cord. Scale bars, 100 μm. **b** The absence of *sec14l3* expression in the trunk vascular system of *etsrp* mutant embryos at 26 hpf. High-resolution views of the boxed regions are shown at right panels. Scale bars, 100 μm. **c** Quantitative RT-PCR of *sec14l3* mRNA in both GFP[+] cells and GFP[-] cells from *Tg(fli1a:EGFP)^{y1}* transgenic embryos at 24 hpf. *sec14l3* is highly enriched in GFP[+] cells (blue bars) relative to negative cells (red bars), using β-actin as the internal control. *kdrl/vegfr2* mRNA serves as a positive control for vascular endothelial cells. A two-tailed *t*-test was used for statistical analysis. *$p < 0.05$, the exact p-values in each figure are shown in the Source data file. **d** SEC14L2 is co-localized with VEGFR2 in HUVECs. Red and green represent anti-VEGFR2 antibody and anti-SEC14L2 antibody staining signal respectively; blue symbolizes DAPI stained nucleus. Regions in the boxes are enlarged in the right corner. Scale bars, 20 μm in the original pictures and 5 μm in the enlarged panels

RT-PCR analysis (Supplementary Fig. 1b). Compared to control embryos injected with standard MO (std-MO), 5 ng sec14l3-tMO2 injection showed no obvious difference in morphology, with normal convergent and extension movements at gastrulation stages (Supplementary Fig. 1c). When analyzing vascular development, we found that knockdown of *sec14l3* in Tg(kdrl: GFP)^{s843Tg} embryos resulted in arterial-venous segregation and luminal expansion defects with compromised ISVs formation (Fig. 2a and Supplementary Fig. 2a). However, when sec14l3-tMO2 was injected in a *p53* mutant background, ISVs sprouting defect was largely restored at 48 hpf (Fig. 2a and Supplementary Fig. 2b), which presumably relieves the general off-target effect of morpholinos[33]. Therefore, we conclude that *sec14l3* specifically regulates DA and PCV formation during vasculogenesis.

To figure out *sec14l3* functions in a maternal- or zygotic-dependent manner, we turned to use the splicing blocker sec14l3-sMO. Consistent with sec14l3-tMO2, sec14l3-sMO injection

similarly inhibited the arterial-venous segregation at 23 hpf and luminal expansion at 25 hpf (Fig. 2b, c), suggesting that the vascular defects mainly arise from the loss of zygotic *sec14l3*. More importantly, the DA and PCV defects in *sec14l3* morphants were restored by *sec14l3* mRNA overexpression (Fig. 2b, c), indicative of specific effects.

To substantiate the knockdown effect, we also generated a *sec14l3* mutant based on clustered regularly interspaced short palindromic repeats (CRISPR)/Cas9 technology, using a guide RNA targeting the 5th exon of *sec14l3* gene in Tg(kdrl:GFP)^{s843Tg} transgenic line (Supplementary Fig. 3a). The isolated mutant allele contains a 10-bp deletion near the target site, causing a premature stop codon and presumably a truncated protein only owning CRAL-TRIO domain (Supplementary Fig. 3a). Although there is no obvious difference between mutant and wild-type embryos in morphology (Supplementary Fig. 3b), *sec14l3* transcripts are almost abolished in homozygous mutants

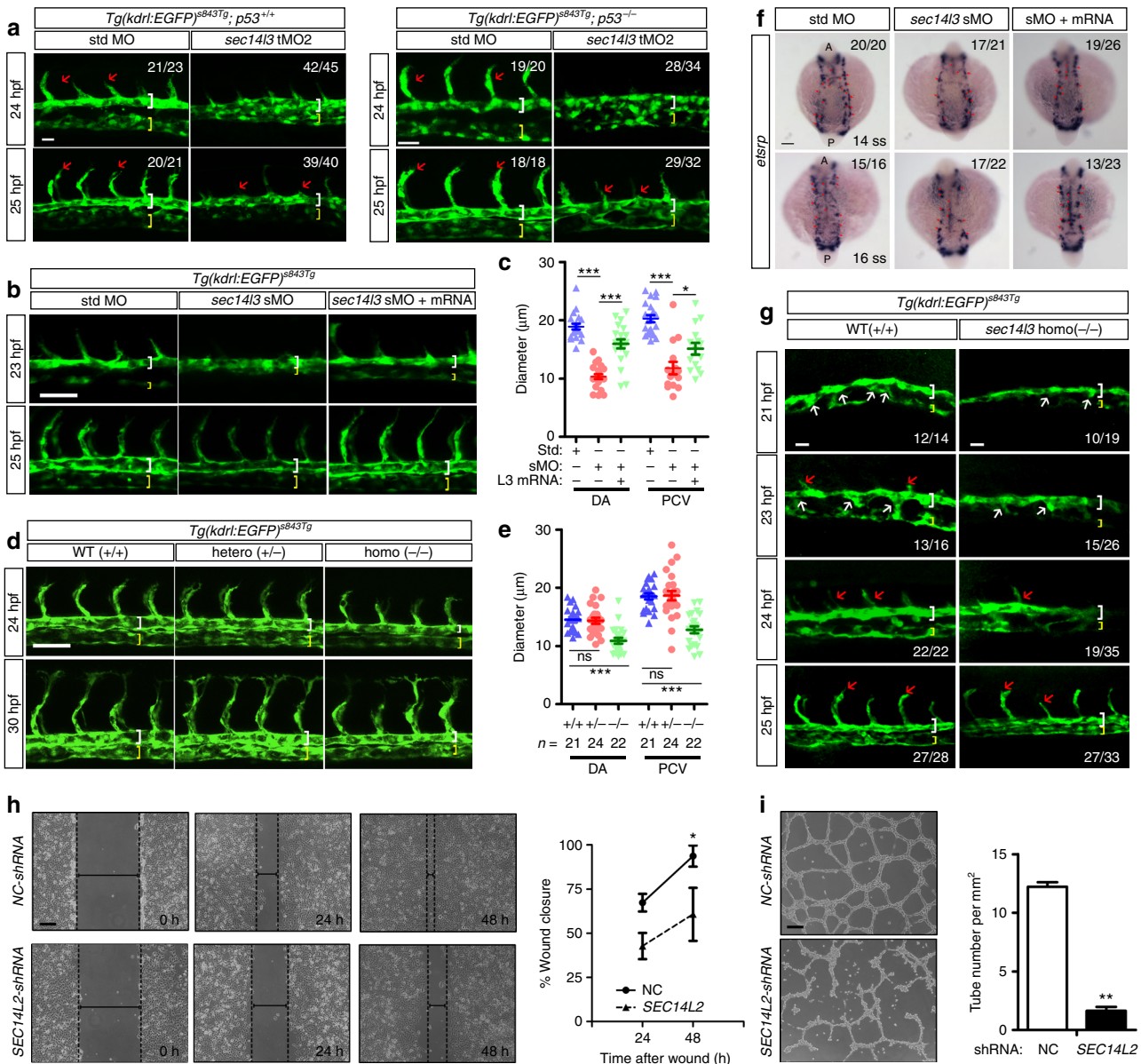

**Fig. 2** *sec14l3/SEC14L2* are required for vascular formation in vivo and in vitro. **a** sec14l3-tMO injection impairs arterial-venous segregation and luminal formation in zebrafish. Embryos of *Tg(kdrl:EGFP)*[s843Tg]*;p53*[+/+] (left panel) or *Tg(kdrl:EGFP)*[s843Tg]*;p53*[−/−] fish (right panel) were used. White and yellow brackets indicate DA and PCV respectively; red arrows indicate ISVs sprouting. The ratio in the right corner indicates the number of embryos with observed pattern/the total number of observed embryos. **b** sec14l3-sMO mediated knockdown of *sec14l3* causes trunk vascular defects. 0.5 ng sec14l3-sMO and 150 pg *sec14l3* mRNA were co-injected for rescue experiment. **c** Statistical analyses of DA and PCV diameters at 25 hpf in **b** (*n* = 20 embryos). **d** Vasculature defects in *sec14l3* mutant embryos. Heterozygous mutants were intercrossed and embryos were harvested for vasculature observation and genotyping analysis individually. **e** Statistical analyses of DA and PCV diameters at 30 hpf in **d**. The total number of embryos in each genotype is indicated. **f** sec14l3-sMO injection causes angioblast migration defects. 0.5 ng sec14l3-sMO and 100 pg *sec14l3* mRNA were co-injected for rescue experiment. WISH using *etsrp* probe marks the medial (arrowhead) and lateral (arrow) progenitor populations simultaneously. **g** Ventral sprouting defects in *sec14l3* mutant embryos. **h** SEC14L2 knockdown inhibits the in vitro wound closure process. HUVECs infected with control or *SEC14L2* shRNA were used for wound closure observation. The ratio of recovered wound width at a specific time point post-wounding to the initial wounding width was calculated. Dashed lines indicate wound edges. Statistical data are shown (*n* = 3). **i** SEC14L2 knockdown inhibits in vitro tube formation. HUVECs were infected with shRNA-packed lentivirus and the complete tube number was scored per field at 24 h after plating. The statistical data are shown on the right (*n* = 3 independent experiments, 60 fields total). All statistical data are shown as mean ± SEM. One-way ANOVA tests were used for statistical analyses in **c**, **e**, and *t*-tests were used for **h–i**. *$p < 0.05$; **$p < 0.01$; ***$p < 0.01$; ns, not significant. Scale bars, 25 μm for **a**, 50 μm for **b**, **d**, **g**, and **h–i**, 100 μm for **f**. Source data are provided as a Source Data file

(Supplementary Fig. 3c). Compared with wild-type and heterozygous, homozygous embryos exhibit arterial-venous segregation and luminal expansion defects with nearly normal ISVs formation, distinct from impaired ISV length in *sec14l3* morphants (Fig. 2d, e, and Supplementary Fig. 4). This is

partially due to a non-specific effect of tMOs, as after injection of sec14l3-tMO2 in homozygotes, it could not cause further DA and PCV defects, but with further reduction of ISV length (Supplementary Fig. 4). Therefore, we conclude that *sec14l3* is required for zebrafish DA and PCV formation.

Since it is reported that two distinct mechanisms could account for the DA and PCV formation in zebrafish[34,35], we wonder what happens in our mutants. Using *fli* and *etsrp* as probes, the WISH result showed that in *sec14l3* deficient embryos, the specification of angioblast is not affected at 8-somite stages (Supplementary Fig. 5), while its migration is significantly reduced at 14–16 somite stages, which could be rescued by the injection of *sec14l3* mRNA (Fig. 2f). Meanwhile, with high-resolution imaging in *Tg(kdrl:GFP)*[s843Tg] embryos, we found that the sprouting of venous progenitors from the vascular cord is also decreased from 21 to 23 hpf in *sec14l3* deficient embryos (Fig. 2g and Supplementary Fig. 6). Therefore, we'd like to conclude that *sec14l3* play important roles in zebrafish DA and PCV blood vessels formation via regulating angioblast migration.

Since *SEC14L2* is expressed in HUVECs (Fig. 1d and Supplementary Fig.1a), we assessed its function by in vitro wound closure assay in HUVECs. Compared with NC-shRNA infected control, the wound gap in *SEC14L2* shRNA cells was closed much slower (Fig. 2h), suggesting a requirement of *SEC14L2* for endothelial cell migration. The in vitro tube formation assay in HUVECs[36] revealed that knockdown of *SEC14L2* obviously inhibited the formation of capillary-like tubes (Fig. 2i), indicating a facilitating effect of *SEC14L2* on the tube formation of endothelial cells. Therefore, *SEC14L2* may mainly promote mammalian endothelial cell migration to maintain vasculature formation.

**Sec14l3/SEC14L2 promote VEGF-induced VEGFR2 activation.** VEGFR2 signaling plays vital roles during zebrafish vasculature, which could be motivated both by VEGF ligands and blood flow-induced fluid shear stress (FSS)[10,37]. Given that SEC14L2 co-localizes with VEGFR2 in HUVECs (Fig. 1d), we hypothesized that SEC14L2 regulates VEGFR2 signaling. Since Sec14l3 exerts its function on angioblast migration during vasculogenesis before the onset of blood flow in vivo (Fig. 2f, g and Supplementary Fig. 6), we focused on the VEGF- but not FSS-induced VEGFR2 signaling transduction in vitro, and further investigated *SEC14L2* knockdown effect in HUVECs on the downstream effectors of VEGF signaling to mimic the endogenous regulation. As shown in Fig. 3a, the addition of VEGFa led to a rapid upregulation of p-ERK-1/2 (Thr202/Tyr204) and p-AKT (Ser473) in NC-shRNA transfected control cells. Strikingly, both of these responses to VEGFa stimulation were markedly reduced in *SEC14L2* shRNA transfected cells (Fig. 3a–c). Additionally, in human umbilical arterial endothelial cells (HUAECs) with *SEC14L2* knockdown, the p-AKT level was also inhibited significantly, whereas the p-ERK level was not changed a lot (Supplementary Fig. 7). In line with the compromised VEGF signaling in HUVECs, depletion of *sec14l3* also caused a reduction of p-Erk and p-Akt in zebrafish embryos (Fig. 3d, e). These results indicate that Sec14l3/SEC14L2 positively regulate VEGF signaling pathway.

Next, we tried to rescue the vasculogenic defects (i.e., narrow DA and PCV) in *sec14l3* morphants by injecting *ca-MEK* and/or *ca-AKT* mRNAs encoding the constitutively active (ca) forms of human MEK and AKT respectively[38,39]. To our surprise, 50 pg *ca-MEK* mRNA injection alone only restored the DA formation in *sec14l3* morphants, while 50 pg *ca-AKT* mRNA only rescued PCV. When the two mRNA species were co-injected, both DA and PCV of morphants were largely restored (Fig. 3f, g). These data provide strong evidence that *sec14l3* acts through VEGF signaling to regulate vascular formation.

**SEC14L2 promotes VEGFR2 phosphorylation at Y$^{1175}$ site.** Upon VEGFa stimulation, VEGFR2 molecules on the plasma membrane undergo dimerization and phosphorylation at several key tyrosine residues including the Y$^{1054/1059}$ and Y$^{1175}$ sites, which are essential for VEGF signaling to activate its downstream PLCγ/ERK and PI3K/AKT pathways[40]. The finding that *SEC14L2* knockdown reduces ERK and AKT activation (Fig. 3a–e, and Supplementary Fig. 7) prompted us to examine phosphorylation levels of VEGFR2 at both Y$^{1054/1059}$ and Y$^{1175}$ sites. Results showed that *SEC14L2* knockdown had no significant effect on VEGFR2 Y$^{1054/1059}$ phosphorylation (Fig. 4a, b). In contrast, Y$^{1175}$ phosphorylation level showed a pronounced reduction in *SEC14L2* shRNA cells (Fig. 4a, c), which is consistent with the reduction of activated ERK and AKT by *SEC14L2* knockdown (Fig. 3a–c). To confirm functional conservation between zebrafish Sec14l3 and human SEC14L2 in VEGF signal transduction, a Flag-tagged full-length form of zebrafish Sec14l3 was transfected into *SEC14L2* knockdown stable HEK293T cells with VEGFR2 overexpression. As a result, *sec14l3* overexpression not only restored but even enhanced VEGFR2-Y$^{1175}$ phosphorylation following VEGFa stimulation (Fig. 4d, e). To test whether this regulation is specific, cytoplasmic regions of three different VEGF receptors, VEGFR1-cyto, VEGFR2-cyto, and VEGFR3-cyto, were overexpressed with Sec14l3 for co-IP experiments respectively in HEK293T cells. Results showed that only VEGFR2-cyto could strongly interact with Sec14l3 (Fig. 5a), indicating that Sec14l3 somehow specifically regulates VEGFR2, but not VEGFR1 or VEGFR3 signaling. Taken together, these data indicate that zebrafish Sec14l3 is the counterpart of human SEC14L2 and both of them enhance VEGF signaling via regulating VEGFR2-Y$^{1175}$ phosphorylation specifically.

Previous studies have demonstrated that decreased phosphorylation of VEGFR2-Y$^{1175}$ may be caused by prolonged exposure of the activated VEGFR2 to some tyrosine phosphatases beneath or on the plasma membrane. PTP1B has been reported and identified as such a specific phosphatase[21,22,41,42]. To prove this notion, we compared the interaction between PTP1B and VEGFR2-cyto with or without SEC14L2 in HEK293T cells. The co-IP result showed that SEC14L2/Sec14l3 could indeed prevent VEGFR2 from interacting with PTP1B (Fig. 5b). Then, we wondered whether knockdown of *PTP1B* could reverse the effect of *SEC14L2* depletion on VEGFR2-Y$^{1175}$ phosphorylation. Strikingly, co-knockdown of *PTP1B* and *SEC14L2* in HEK293T cells resulted in much higher phosphorylation levels of VEGFR2-Y$^{1175}$ compared to *SEC14L2* knockdown alone (Fig. 5c, d).

Consistent with these in vitro results, overexpression of *VEGFR2-cyto* mRNA or co-knockdown of *ptp1b* in *sec14l3* morphants led to a relatively normal lumen size of DA and PCV at 25 and 30 hpf (Fig. 5e–h), while overexpression of *VEGFR2-cyto* mRNA or knockdown of *ptp1b* alone had no obvious effect on the DA and PCV formation (Supplementary Figs. 8, 9). Taken together, these in vitro and in vivo observations imply that SEC14L2/Sec14l3 may somehow protect VEGFR2-Y$^{1175}$ phosphorylation from its specific phosphatase PTP1B and then stimulate VEGF signaling.

**VEGFR2 trafficking is suppressed after *SEC14L2* depletion.** Based on the above results, in particular, the vesicle-like co-localization of SEC14L2 and VEGFR2 in HUVECs (Fig. 1d), we hypothesize that Sec14l3/SEC14L2 may contribute to intracellular trafficking of VEGFR2. To address this issue, we performed surface biotinylation assay to detect internalization and recycling of overexpressed VEGFR2 in HEK293T cells (Supplementary Fig. 10). We found that, before induction of internalization, the biotin-labeled VEGFR2 protein level on the cell surface in *SEC14L2* shRNA transfected HEK293T cells was comparable

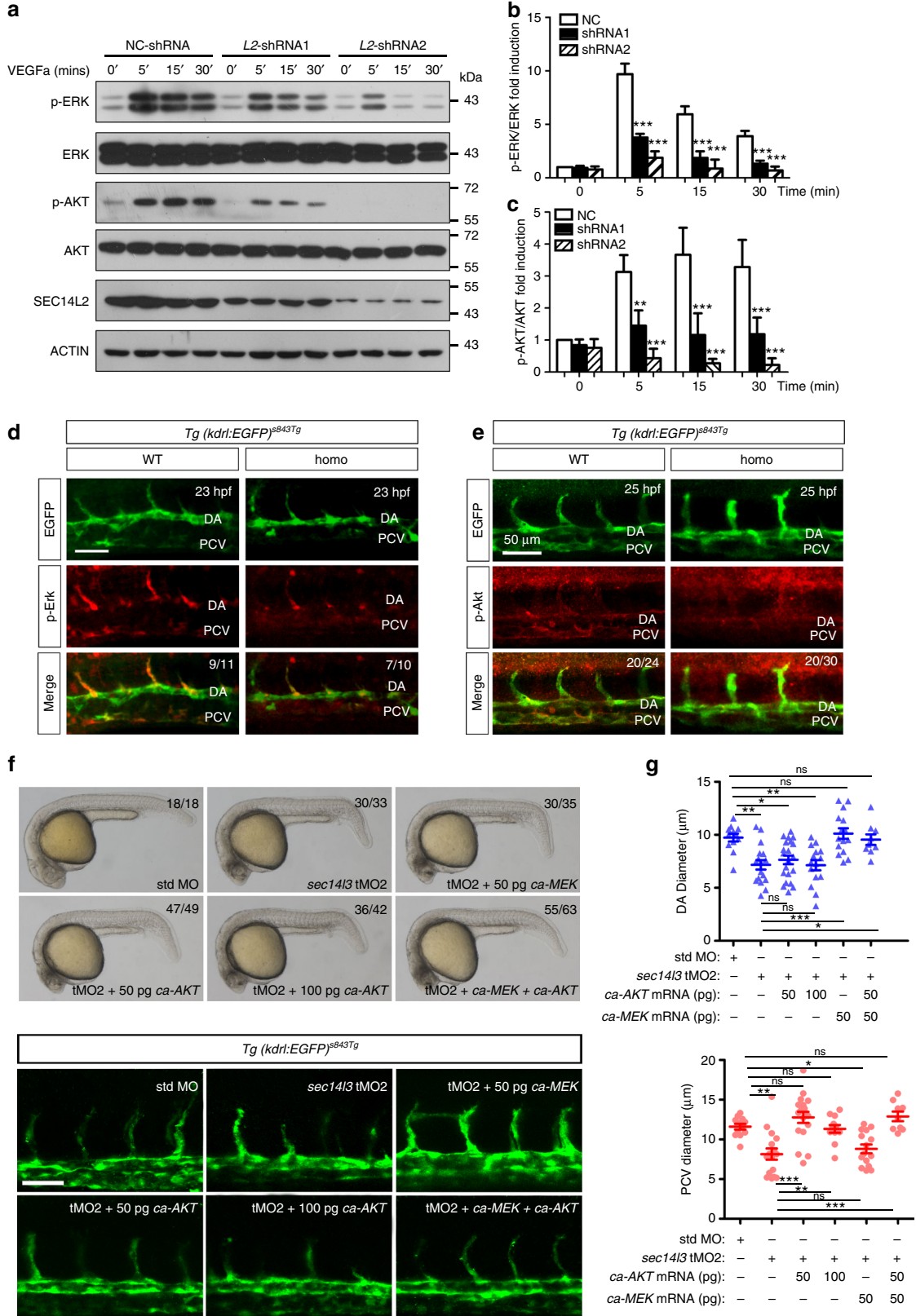

to that in control cells (Fig. 6a, the first lane and fifth lane); following induction of internalization, intracellular biotin-labeled VEGFR2 proteins gradually increased in the control cells, but this increase was much slower in *SEC14L2* shRNA cells (Fig. 6a). Thus, SEC14L2 is a potential regulator of VEGFR2 trafficking.

Next, the transport of VEGFR2-containing endosomes in HUVECs was examined by immunostaining assay. Before induction of internalization, endogenous VEGFR2 proteins on the cell surface were labeled similarly in control and *SEC14L2* shRNA transfected cells (Fig. 6b, the first panel). After internalization induction at 37 °C for 10 min, the

**Fig. 3** *SEC14L2/sec14l3* knockdown attenuates VEGF signaling. **a** *SEC14L2* knockdown counteracts VEGFa-motivated p-ERK and p-AKT levels in HUVECs. HUVECs were infected with NC or *SEC14L2* shRNA for 48 h. After VEGFa stimulation for 5, 15 or 30 min, cell lysates were harvested and immunoblotted with indicated antibodies. **b**, **c** Statistical results of relative p-ERK (**b**) and p-AKT (**c**) levels in **a**. The grey intensity of each band was measured for calculating the ratio of p-ERK to ERK (**b**) and p-AKT to AKT (**c**). Data are then normalized to control group with 0′ stimulation and represented as mean ± SEM from three independent experiments. **d**–**e** Depletion of *sec14l3* decreases p-Erk (**d**) and p-Akt (**e**) levels in zebrafish embryos. Embryos from intercrossing *sec14l3* heterzygous in *Tg(kdrl: GFP)[s843Tg]* transgenic background were harvested at 23 hpf for p-Erk antibody immunostaining (**d**) or at 25 hpf for p-Akt antibody staining (**e**). DA, dorsal aorta. The ratio in the right corner indicates the number of embryos with reduced staining/the number of observed embryos. Scale bars, 50 μm. **f** The defective DA/PCV lumen formation in *sec14l3* morphants can be partially rescued by *ca-MEK* or *ca-AKT* mRNA injection. Five nanograms std-MO or sec14l3-tMO2 in combination with 50 or 100 pg *ca-MEK* or *ca-AKT* mRNA was injected into one-cell stage embryos. Their morphology (the upper panel) or vasculature defects (the bottom panel) at 25 hpf are separately shown. The ratio in the right corner indicates the number of embryos with indicated morphology/the number of observed embryos. Scale bars, 50 μm. **g** Statistic results of the DA (upper panel) or PCV (bottom panel) luminal diameters of each group at 30 hpf. Twenty embryos of each group from one representative experiment were calculated and shown here. To quantify the DA and PCV luminal diameter for an embryo, five different vessel regions of the same fish were measured to calculate the mean value, which was used to represent its vessel diameter of this fish. Three independent experiments were carried out. *$p < 0.05$; **$p < 0.01$; ***$p < 0.001$; ns, not significant. ANOVA tests were used for statistical analyses. Source data are provided as a Source Data file

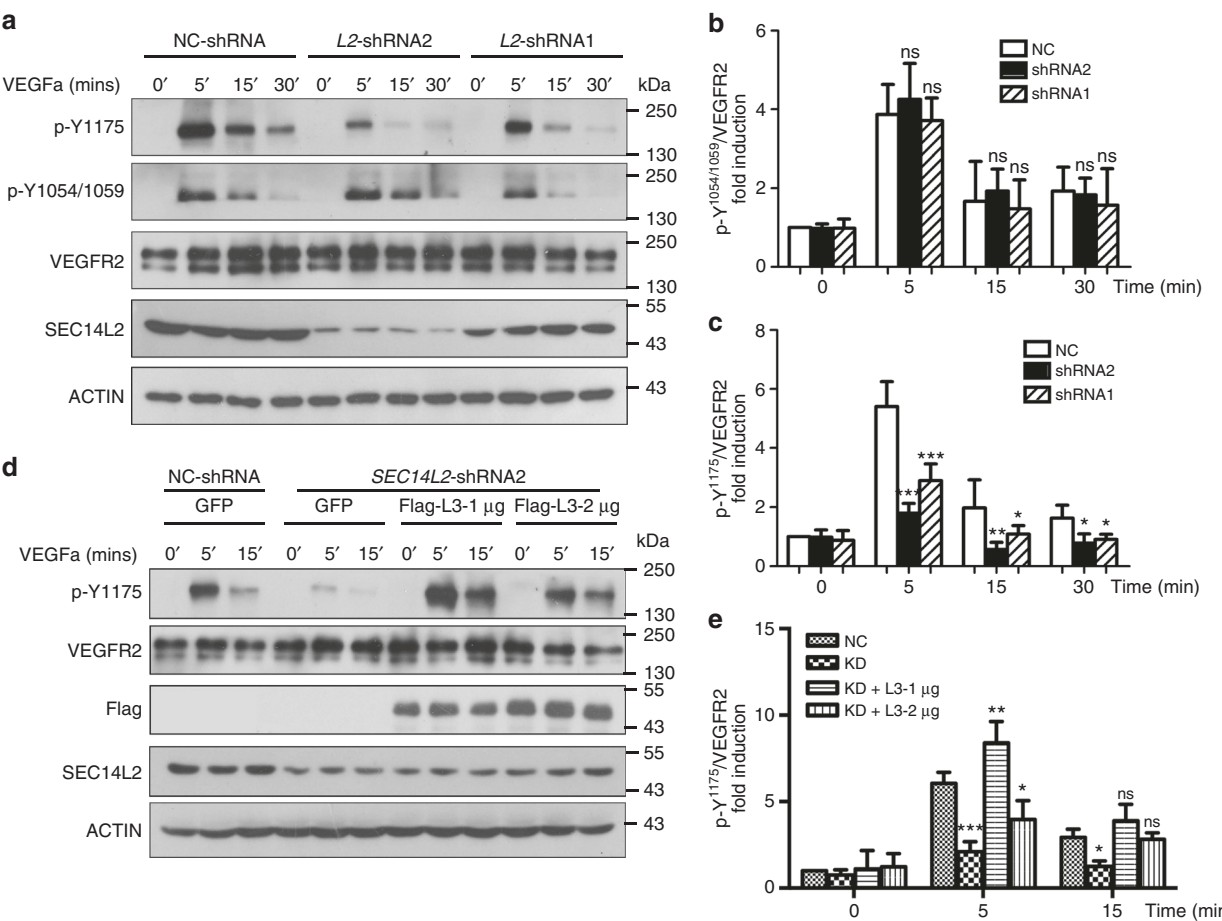

**Fig. 4** *SEC14L2* knockdown decreased VEGFR2 phosphorylation at Y[1175] site. **a** Western blot analysis of cell lysates from NC or *SEC14L2* shRNA infected HUVECs. Cells were infected and starved overnight, followed by 100 ng ml[-1] VEGFa stimulation for 5, 15 or 30 min, respectively. VEGFR2 (p-Y[1175]), VEGFR2 (p-Y[1054/1059]), and total VEGFR2 levels were examined by immunoblotting using respective antibodies. **b**, **c** Statistical results of p-VEGFR2-Y[1054/1059] (**b**) and p-VEGFR2-Y[1175] (**c**) levels in **a** relative to total VEGFR2. Data are normalized to control group without stimulation and shown as mean ± SEM from three independent experiments. *$p < 0.05$; **$p < 0.01$; ***$p < 0.001$; ns, not significant. **d**, **e** Overexpression of zebrafish *sec14l3* in HEK293T cells restores the level of p-VEGFR2-Y[1175] inhibited by *SEC14L2* knockdown. One or two micrograms sec14l3 plasmid was transfected into *SEC14L2* knockdown stable HEK293T cells with VEGFR2 overexpression. Following starvation overnight, 100 ng ml[-1] VEGFa was added and stimulation continues for 0, 5 or 15 min. Quantification data of the relative p-VEGFR2-Y[1175] level from three repeated experiments are normalized to control group with 0′ stimulation and shown on the right (**e**). *$p < 0.05$; **$p < 0.01$; ***$p < 0.001$; ns, not significant. The loading volume of **a**, **d** was adjusted for the same amount of total VEGFR2 level in each lane. Two-way ANOVA tests were used for statistical analyses in **b**, **c**, and **e**. Source data are provided as a Source Data file

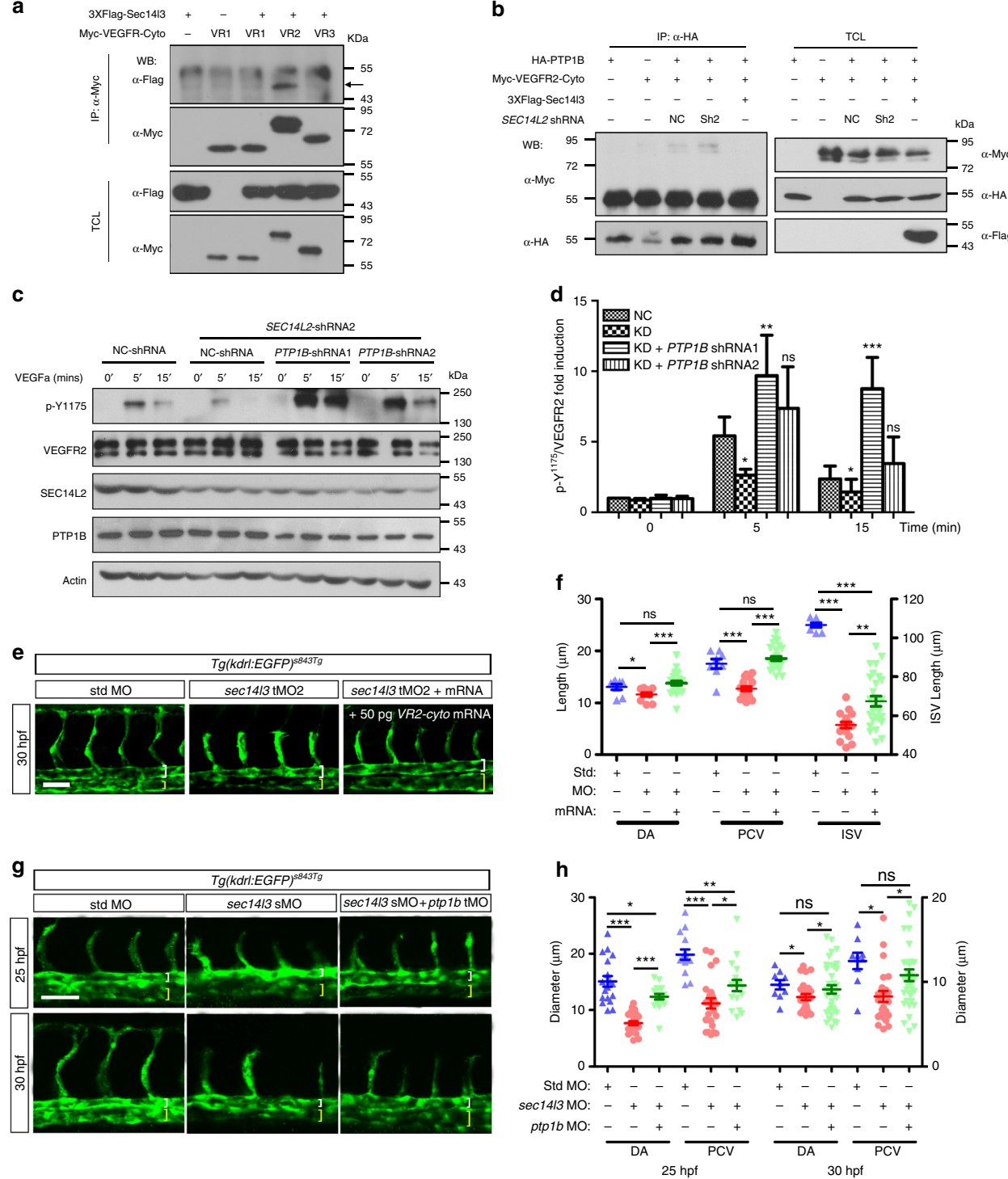

VEGFR2-containing endosomes in *SEC14L2* knockdown HUVECs were significantly decreased compared to the control cells (Fig. 6b, c). Additionally, co-localization analysis between internalized VEGFR2 and the early endosomal marker EEA1 showed that *SEC14L2* shRNA cells contained far fewer VEGFR2/EEA1 double positive endosomes than control cells (Fig. 6b, c), suggesting a requirement of SEC14L2 for VEGFR2 internalization into early endosomes.

It has been reported that in resting endothelial cells, VEGFR2 displays both the cell surface distribution and internal vesicular pool storage, which can be redistributed to the cell periphery upon VEGF stimulation[12]. To investigate whether SEC14L2 participates in this process, we performed surface biotinylation assay over a 35-min time course (20 min internalization at 22 °C followed by 15 min recycling at 37 °C). In NC-shRNA control cells, about 40% of internalized VEGFR2 could be recycled back to the plasma membrane and this recycling rate was enhanced to approximately 80% in the present of VEGFa stimulation (Fig. 6d), which is similar to the previous report[12,22]. However, knocking down *SEC14L2* drastically decreased the VEGFR2 recycling rate to about 8 and 20%, respectively, with or without VEGFa stimulation (Fig. 6d). Therefore, we conclude that SEC14L2 may promote VEGFR2 intracellular mobilization at least in two ways, internalization from and recycling back to the cell surface.

**Fig. 5** Enhanced VEGFR2-Y$^{1175}$ phosphorylation restores *SEC14L2/sec14l3* depletion effects. **a** Sec14l3 specifically interacts with the cytoplasmic region of VEGFR2 in HEK293T cells. The cytoplasmic region of VEGFR1, VEGFR2 or VEGFR3 was individually transfected with Flag-tagged Sec14l3 into HEK293T cells for immunoprecipitation (IP) and immunoblotting. TCL, total cell lysate. **b** SEC14L2/Sec14l3 could alleviate the interaction between PTP1B and the VEGFR2-cyto in HEK293T cells. HA-tagged PTP1B and Myc-tagged VEGFR2-cyto were co-transfected into HEK293T cells with Flag-tagged Sec14l3 or *SEC14L2* shRNA. After 72 hpf, cells were harvested and lysed for immunoprecipitation assay using an HA antibody. WB, western blot. **c, d** *PTP1B* knockdown rescues p-VEGFR2-Y$^{1175}$ level inhibited by *SEC14L2* knockdown in HEK293T cells. *PTP1B* shRNA1 or shRNA2 was transfected into *SEC14L2* shRNA stable HEK293T cells with VEGFR2 overexpression. p-VEGFR2-Y$^{1175}$ levels were checked after 100 ng ml$^{-1}$ VEGFa stimulation for 0, 5 or 15 min. The loading volume was adjusted for the same amount of total VEGFR2 level in each lane. Quantification data of the relative p-VEGFR2-Y$^{1175}$ level from three independent experiments are shown in **d** ($n = 3$). **e** The cytoplasmic region of *VEGFR2* mRNA rescues luminal defects in *sec14l3* morphants. Five nanograms std-MO or sec14l3-tMO2 in combination with 50 pg *VEGFR2-cyto* mRNA was injected into one-cell stage embryos from *Tg(kdrl: GFP)$^{s843Tg}$* transgenic fish, and harvested at 30 hpf for the measurement of luminal diameters /ISV length. **f** Statistical result of the luminal diameters of DA, PCV, and ISV in **e**. 20 embryos of each group from one representative experiment were calculated and shown here. Three independent experiments were carried out. **g, h** ptp1b knockdown partially restores the lumen size of DA/PCV in zebrafish *sec14l3* morphants. *Tg(kdrl:GFP)$^{s843Tg}$* transgenic embryos were injected with 0.5 ng std-MO or sec14l3-sMO in combination with 5 ng ptp1b-tMO and harvested at 25 and 30 hpf for the measurement of luminal diameters. Twenty embryos of each group were calculated, and statistical data from three independent experiments are shown in **h**. *$p < 0.05$; **$p < 0.01$; ***$p < 0.001$; ns, not significant. ANOVA tests were used for statistical analyses in **d**, **f**, and **h**. Scale bars, 50 μm. Source data are provided as a Source Data file

Furthermore, we tried to figure out the final destination of VEGFR2 after knocking down *SEC14L2* in HUVECs. Western blotting results showed that the total VEGFR2 protein level is not significantly affected when SEC14L2 is depleted, exclusive of its possible role in VEGFR2 degradation (Supplementary Fig. 11). When co-staining VEGFR2 with late endosome markers, RAB7 or CD63, we found that SEC14L2 knockdown led to more VEGFR2 accumulation in these compartments (Fig. 6e, f). Taken these results together, we would like to prospect that more VEGFR2 is stored in late endosomes but not delivered to the lysosome for degradation in SEC14L2 deficient cells, which is consistent with previous reports[12].

**Sec14l3/SEC14L2 physically interact with VEGFR2 and RABs.** We next set out to investigate how Sec14l3/SEC14L2 mediate VEGFR2 trafficking. Given that Sec14l3 interacts with VEGFR2-cyto specifically (Fig. 5a) and SEC14L2 co-localizes with VEGFR2 endogenously in vesicular compartments (Fig. 1d), we first demonstrated that endogenous SEC14L2 indeed interacts with endogenous VEGFR2 and this interaction could be strengthened upon VEGFa stimulation (Fig. 7a), indicating that SEC14L2 prefers binding to activated VEGFR2. To further prove this point, wild-type form (WT-VEGFR2) or kinase mutation form (KM-VEGFR2) of VEGFR2 was co-transfected with Flag-tagged zebrafish Sec14l3 in HEK293T cells for Co-IP experiments[43]. We found that Sec14l3 associates with WT-VEGFR2 much stronger than KM-VEGFR2 (Fig. 7b, c). Taking these data together, we conclude that Sec14l3 favors associating with activated VEGFR2. Additionally, domain mapping results revealed that the Sec14 domain of Sec14l3 was essential for its interaction with VEGFR2 (Fig. 7d, e).

As a member of the receptor tyrosine kinases (RTKs) family, VEGFR2 is regulated by various Rab proteins at the intracellular trafficking level, which renders vesicle transport to particular destination somehow through recruiting their corresponding effectors[44]. Considering that Sec14l3/SEC14L2 act during VEGFR2 trafficking, we wondered whether Sec14l3 could associate with RAB family proteins. Four members of RAB proteins, representing distinct endocytosis routes[45–47], were selected for co-IP experiments in HEK293T cells. Results revealed that only RAB4A and RAB5A could strongly interact with Sec14l3 (Fig. 7f), which coincides with the function of Sec14l3 in VEGFR2 internalization and recycling (Fig. 6). Furthermore, these interactions were confirmed by endogenous co-IP data in HUVECs (Fig. 7g, h). Domain mapping analyses disclosed that the N-terminal CRAL-TRIO domain and the C-terminal GOLD2 domain of Sec14l3 were required for interaction with RAB4A

(Fig. 7d, i). Therefore, Sec14l3 may promote VEGFR2 endocytosis and recycling through its interaction with RAB5A and RAB4A.

**Sec14l3/SEC14L2 are required for RAB5A/4A activation.** Like other GTPases, the RAB5A/4A cycle between an active (GTP-bound) and an inactive (GDP-bound) state, which is of critical importance to modulate vesicle trafficking processes[48,49]. We wonder whether Sec14l3/SEC14L2 are required for the activity of RAB5A/4A to regulate VEGFR2 trafficking. To this end, we first compared the association of Sec14l3 with RAB5A/4A-GDP and RAB5A/4A-GTP. Transfection and in vitro binding assays revealed that Sec14l3 preferred binding to the GDP-bound state of RAB5A/4A (Fig. 8a, b). To further investigate the function of Sec14l3/SEC14L2 in RAB5A/4A regulation, we performed immunoprecipitation assay using a specific anti-RAB5A/4A-GTP antibody or pull-down assay using purified GST-R5BD, which specifically binds to RAB5A-GTP[50]. Compared to the NC-shRNA control, knocking down *SEC14L2* caused more than 50% reduction of the RAB5A-GTP and RAB4A-GTP amount (Fig. 8c, d), which is consistent with the compromised VEGFR2 internalization and recycling processes (Fig. 6). Taken together, these results suggest that Sec14l3/SEC14L2 facilitate the formation of RAB5A/4A in their GTP-bound active states, and then probably accelerate the movement of VEGFR2 vesicles to early endosomes and recycling endosomes to enhance VEGF signaling (Fig. 8e).

## Discussion

Given the critical roles of endocytosis in VEGFR2 signaling, numerous relevant factors that have effects on receptor trafficking along endosomes have been identified, and most of them were demonstrated to participate in VEGFR2 internalization or degradation[16,22,23,41,51–54]. However, few factors were found to regulate a relatively complete VEGFR2 intracellular trafficking process, such as receptors internalization from and recycling back to the plasma membrane for enhancing signaling output. Here, we propose that the lipid binding proteins Sec14l3/SEC14L2 concurrently provoke VEGFR2 uptake from the plasma membrane and return from the endosomal pool to maximize signaling strength upon ligand stimulation. Mechanistically, Sec14l3/SEC14L2 conjugate activated VEGFR2 into RAB5-positive sorting endosomes and RAB4-positive fast recycling endosomes through their interaction to protect VEGFR2 from dephosphorylation by the peri-membrane tyrosine phosphatase PTP1B (Fig. 8e). Therefore, Sec14l3/SEC14L2 deficiency disturbs VEGFR2 internalization and recycling, leading to a prolonged exposure to PTP1B and subsequent compromises of PLCγ/ERK and PI3K/AKT activation, which consequently results in defective

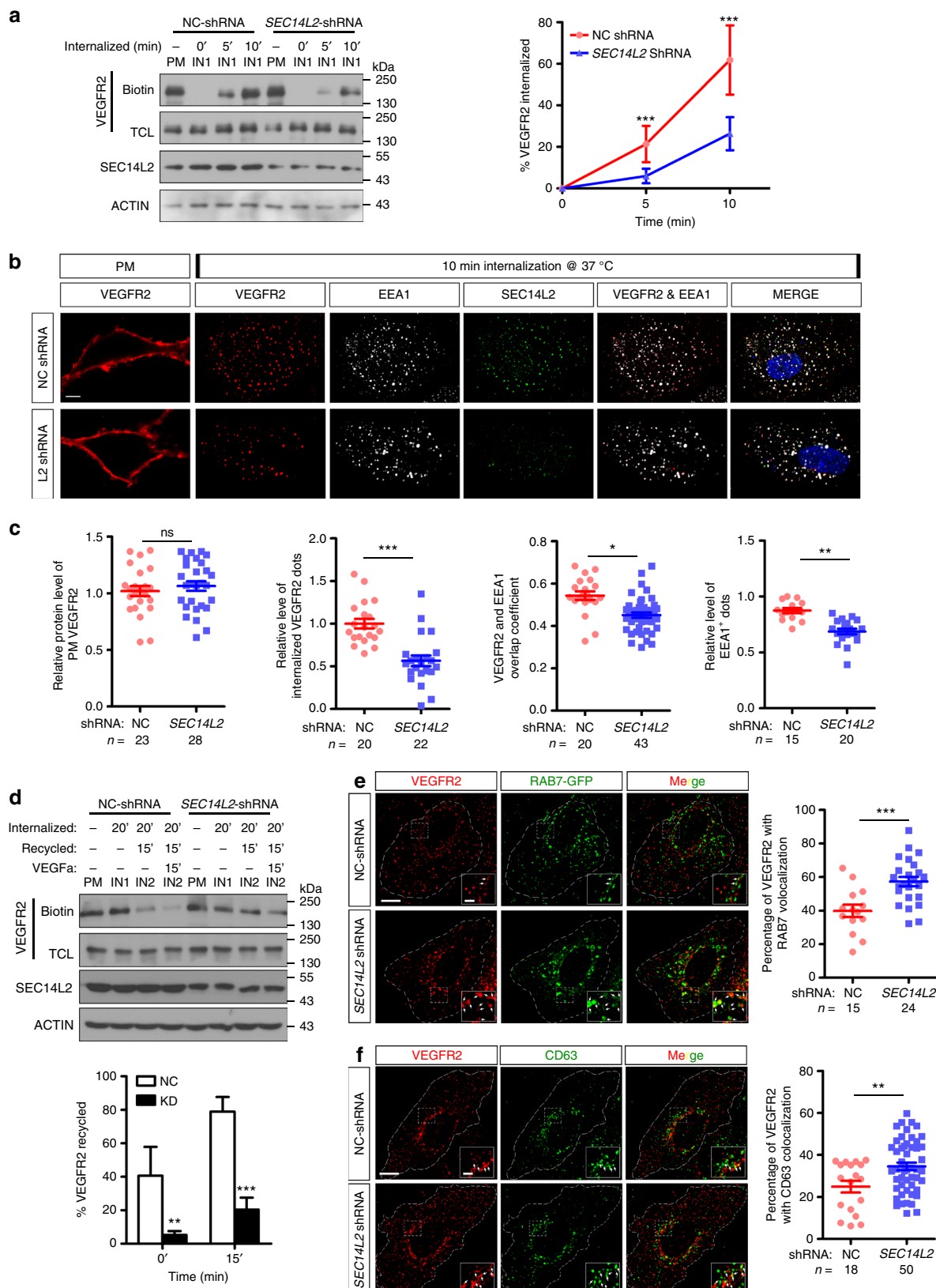

vasculature morphogenesis in zebrafish embryos in vivo and vascular tubulogenesis in vitro. The effective induction of VEGF signaling through overexpression of cytoplasmic region of *VEGFR2* mRNA or suppression of *ptp1b* expression or overexpression of constitutively active *MEK/AKT* partially restored the narrowed DA and PCV lumens in zebrafish *sec14l3*

morphants. This study provides insight into the regulation of VEGFR2 endocytosis in vertebrate vascular system formation.

It was apparent that *sec14l3* inactivation specifically affected vasculogenesis, whereas the angiogenic process appeared to be nearly intact, although both processes heavily rely on VEGF signaling. The pattern of how *sec14l3* is spatiotemporally

**Fig. 6** *SEC14L2* knockdown disturbs VEGFR2 internalization and recycling processes. **a** Western blotting results show impairment of biotinylated VEGFR2 internalization in *SEC14L2* knockdown cells. VEGFR2-overexpressing HEK293T cells were transfected with NC or *SEC14L2* shRNA for internalization assay described as supplementary Fig. 10. Quantified internalized VEGFR2 levels are shown on the right as mean ± SD from three independent experiments (n = 3). Red and blue lines represent control and *SEC14L2* shRNA transfected group, respectively. TCL, total cell lysate. ***$p < 0.001$. **b**, **c** Analysis of VEGFR2 internalization based on immunostaining assay in HUVECs. HUVEC cells infected with NC or *SEC14L2* shRNA were cultured for internalization assay and then cells were fixed for immunostaining with anti-EEA1 (white) and anti-SEC14L2 (green) antibodies. Quantification data of plasma membrane VEGFR2 (red) before internalization, internalized VEGFR2 after internalization, overlapping coefficient between internalized VEGFR2 and EEA1 (n = 40 cells) as well as EEA1 positive dots are shown individually in **c**. *$p < 0.05$; **$p < 0.01$; ***$p < 0.001$; ns, not significant. **d** Western blot analysis of VEGFR2 recycling over a 35 min time course. VEGFR2-overexpressing HEK293T cells were transfected with NC- or *SEC14L2* shRNA for 48 h and used for recycling assay as detailed in Supplementary Fig. 10. The recycling rate of VEGFR2 was measured with or without 100 ng ml$^{-1}$ VEGFa stimulation. Quantification data are shown as mean ± SD from three independent experiments on the right (n = 3). **$p < 0.01$; ***$p < 0.001$. **e**, **f** *SEC14L2* knockdown promotes VEGFR2 localization in RAB7$^+$ (**e**) or CD63$^+$ (**f**) compartments. HUVEC cells infected with NC or *SEC14L2* shRNA were transfected with RAB7-EGFP (**e**) or harvested directly (**f**) for anti-VEGFR2 (red) and EGFP/CD63 (green) staining. Regions in the boxes are enlarged in the right corner. Scale bars, 10 μm. The percentage of VEGFR2 co-localized with RAB7/CD63 is shown individually in the right, n indicates observed cell number. **$p < 0.01$; ***$p < 0.001$. Two-way ANOVA tests were used for **a**, **d** and *t*-tests for **c**, **e**, and **f**. Source data are provided as a Source Data file

expressed may provide an explanation for this. Before the onset of the major axial vessels at 19 hpf, *sec14l3* is expressed in both arterial and venous progenitors. And thereafter, it is gradually restricted into the PCV compartment, not DA or ISVs. This particular time window is corresponding to vasculogenesis, but not angiogenesis context during vascular morphogenesis. We did notice a reduced p-Erk level in DA of *sec14l3* deficient embryos yet with the robust signal in selected endothelial cells, which might be enough for the tip cell selection and subsequent ISVs formation. This notion is compatible with ISV defects in *kdrl* and *plcγ1* mutant embryos, which are caused by a loss of pErk-positive sprouting ISV endothelial cells in dorsal aorta[55].

Inhibition between PI3K/AKT and PLCγ/ERK signaling has been well documented[38,56]. Within the arterial-fated angioblast, VEGFR2 signaling induces PLCγ/ERK kinase cascade and subsequently stimulates expression of several genes in the Notch pathway to ensure the arterial cell fate. It has been reported that overexpression of constitutively active MEK or treatment with agonists of PLCγ/ERK signaling can rescue the DA formation in zebrafish embryos lacking *vegfa*[55,57–60]. On the other hand, in the venous-fated angioblast, PI3K/AKT signaling is activated to antagonize ERK activity, and overexpression of constitutively active Akt promotes the venous cell fate[38]. Therefore, it seems coincident with previous reports that overexpression of *ca-MEK* or *ca-AKT* could rescue the DA or PCV formation respectively in *sec14l3* mutant embryos bearing a relatively low level of VEGF signaling. Since both PLCγ/ERK and PI3K/AKT pathways are downstream branches of the VEGF receptor, how they are preferentially activated in different endothelial progenitor cells or at different developmental stages remain intriguing questions. Furthermore, how Sec14l3 exactly regulates these processes also need to be investigated in the future.

Upon VEGF ligands stimulation, VEGFR2 is engaged in dimerization to induce phosphorylation of several tyrosine residues in its cytoplasmic domain. Meanwhile, various phosphatases have been reported to remove tyrosine phosphorylation at distinct sites[21,61–63]. Given that PTP1B specifically dephosphorylates VEGFR2 at the Y$^{1175}$ site, we merely examined the rescue effect of PTP1B depletion following *Sec14l3/SEC14L2* knockdown. However, we cannot exclude the possible involvements of other phosphatases, especially those transmembrane receptor-type PTPs, such as DEP1/CD148, VE-PTP and so on[62,63]. Knocking down these phosphatases respectively in *Sec14l3/SEC14L2* deficient embryos may provide more information.

Besides VEGF ligands, PECAM-1 mediated fluid shear stress signaling could also recruit VEGFR2 and then phosphorylate it at Y$^{1175}$ site along with VE-Cadherin, thus regulating vascular remodeling, vascular homeostasis, cardiac development and atherogenesis[37]. In this study, we didn't focus on this branch,

largely due to our in vivo observations that *sec14l3* specifically exists and functions during zebrafish vasculogenesis before the onset of blood flow. Additionally, different from mammalian vascular development, it has been reported that changes in shear stress appear to be relatively unimportant in the initial stages of zebrafish angiogenesis because blood vessels can be successfully formed in the absence of heartbeat and blood flow until 14 days postfertilization[64]. Even so, we still cannot exclude that Sec14l3 could participate in regulating fluid shear stress-induced VEGFR2 activation during later stages of zebrafish angiogenesis. It will be interesting to explore this possibility in the future.

Previous studies have shown that VEGFR2 endocytosis proceeds in a clathrin-dependent internalization, followed by the cascade of RAB recruitments to deliver VEGFR2 to the plasma membrane for recycling, into late endosome for storage, or into lysosome for degradation[65]. In this work, we demonstrate that Sec14l3 could directly interact with RAB5A/4A and promote their GTP-bound states formation, and probably orchestrate VEGFR2 internalization and recycling back to the plasma membrane. After performing in vitro GEF assay based on mant-GTPγS fluorescence, we proposed that Sec14l3 could not act as a GEF to accelerate the GTP loading activity of RAB5 directly (Supplementary Fig. 12). Actually, whether Sec14l3 could regulate the GTP hydrolysis activity of RAB5A/4A also need to be determined by MESG-based single-turnover assay. In addition, whether the intrinsic GTPase activity of Sec14l3 is required for the activation of RAB5A/4A also awaits further investigation. Based on our data here, we would like to propose the following potential mechanisms of SEC14L2/Sec14l3 acting on RAB5A/4A activation. (1) Sec14l3 might serve as an adaptor to recruit a GEF protein for RAB5A/4A. Although we have not detected any interaction between Sec14l3 and several well-known GEF proteins of RAB5, such as Rabex5 and RABGEF1, we still cannot exclude this possibility, since there might be other unidentified GEFs for RAB5A/4A that could participate in this process. (2) Sec14l3 belongs to atypical class III PITPs, which are implicated in the traffic of phosphoinositides (PIs) between different membrane compartments[24]. Different PI species display distinct subcellular distributions where they play intriguing functions. For example, PI(3)$P$, PI(4)$P$, and PI(3,5)$P_2$ are found predominantly within early, recycling and late endosomes, respectively[66], which are essential for appropriate endosomal trafficking events. *Drosophila* PI 3-phosphatase MTMR13 promotes PI(3)$P$ turnover at endosomes to activate Rab5 family member Rab21, PI(3)$P$ levels in early endosome may drive RAB5-RAB7 conversion and so on[67,68]. Therefore, it deserves further exploration of whether Sec14l3 could also motivate specific phosphoinositides transportation along vesicles to regulate endosome dynamics, either in morphology or in effector recruitment.

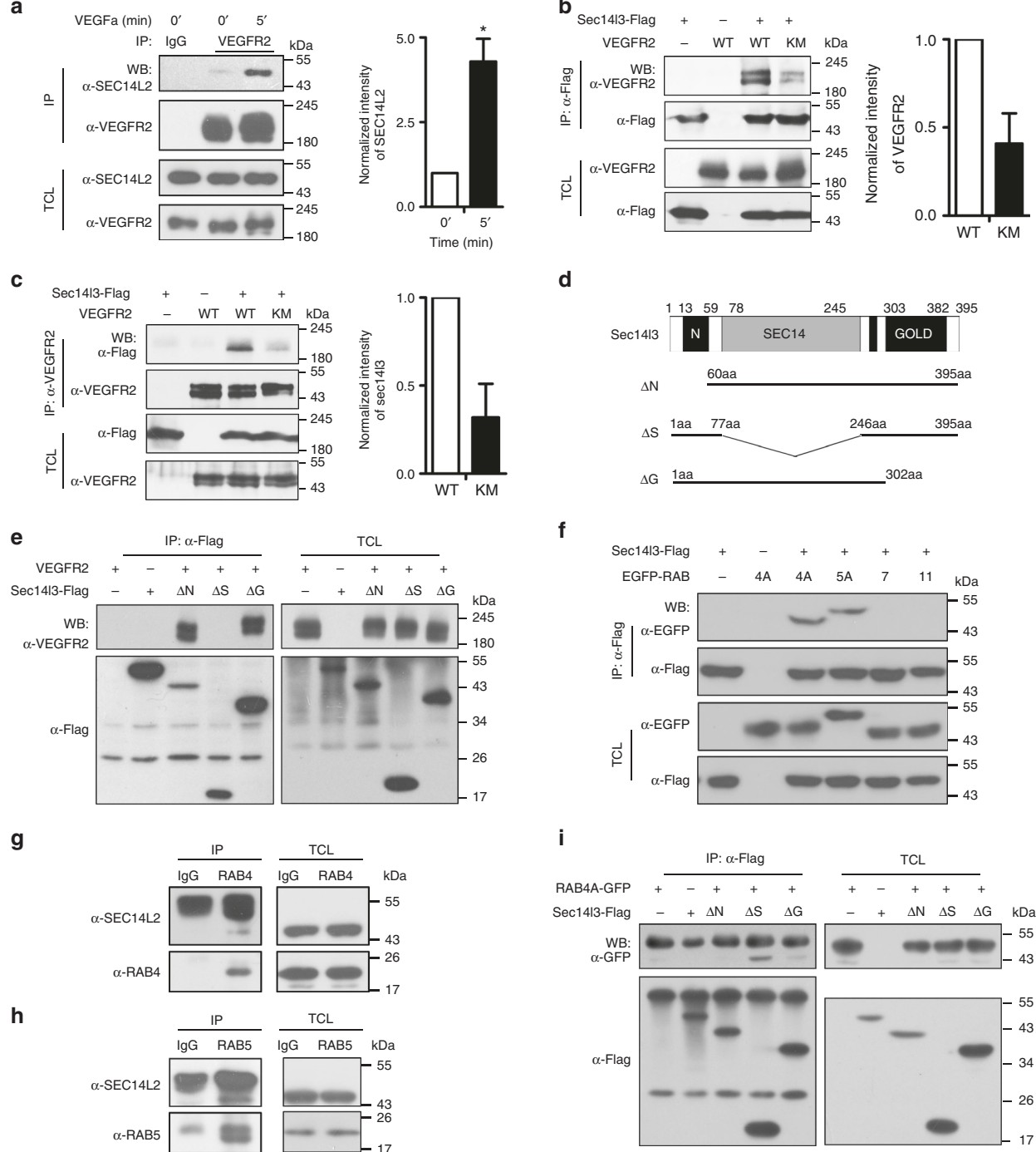

**Fig. 7** Sec14l3/SEC14L2 interact with VEGFR2 and RAB4A/5 A via complementary domains. **a** SEC14L2 interacts with VEGFR2 endogenously in HUVECs. HUVEC cells were starved and stimulated with 100 ng ml⁻¹ VEGFa for 5 min before harvest for IP and immunoblotting. TCL, total cell lysate. IgG serves as a negative control. Quantification of the interaction from three independent experiments is shown on the right (*n* = 3). *p < 0.05. **b**, **c** Sec14l3 shows a much stronger association with WT-VEGFR2 than its kinase domain mutation form (KM-VEGFR2). Different forms of VEGFR2 were co-expressed with Flag-tagged Sec14l3 in HEK293T cells. WB, western blot. Quantification of the interaction is shown on the right (*n* = 3). *t*-tests were used for statistical analyses in **a**–**c**. **d** Schematic diagrams of Sec14l3 and its truncated mutants. Numbers above the diagram indicate the corresponding amino acid positions. ΔN indicates N-terminal CARL-TRIO domain deletion; ΔS, Sec14 domain deletion; ΔG, GOLD domain deletion. **e** Sec14l3 interacts with VEGFR2 via its Sec14 domain. HEK293T cells were transfected with different Sec14l3 domain deletion forms with VEGFR2 plasmid respectively and harvested for IP. **f** Sec14l3 specifically interacts with RAB4A and RAB5A. Different RAB plasmids were co-transfected with VEGFR2 into HEK293T cells respectively for co-IP assay. **g**, **h** SEC14L2 interacts with RAB4 (**g**) and RAB5 (**h**) endogenously in HUVECs. **i** Sec14l3 interacts with RAB4A via its CRAL-TRIO and GOLD2 domains. Source data are provided as a Source Data file

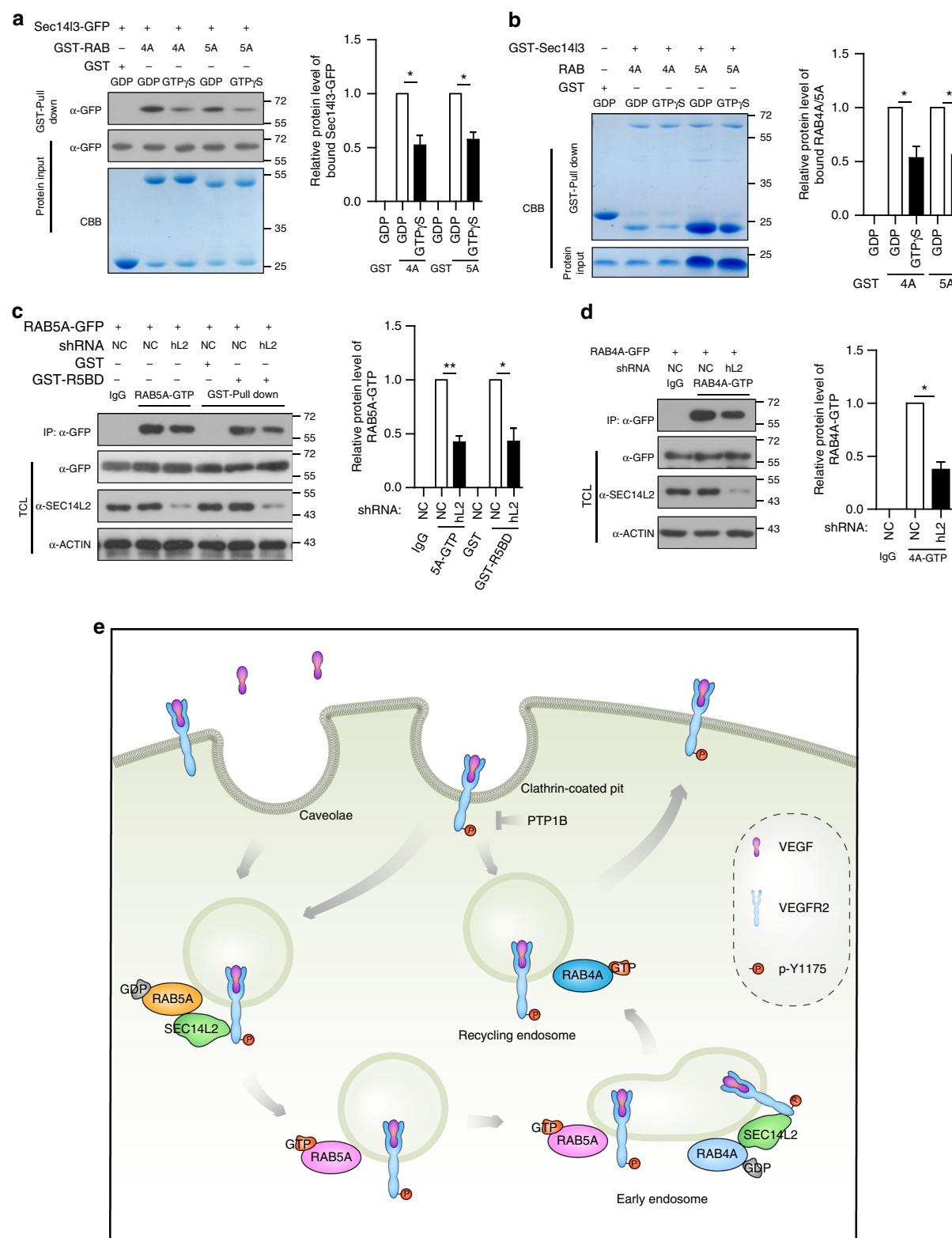

## Methods

**Zebrafish strains and embryos manipulation.** Tuebingen strain of zebrafish (*Danio rerio*) was used. Unless otherwise stated, *Tg(kdrl:GFP)^s843Tg* and *Tg(fli1a: EGFP)^y1* transgenic lines were used for easy observation of vasculature development. Ethical approval is acquired from Tsinghua University Animal Care and Use Committee for the fish maintenance and manipulations, according to the institutional animal care and use committee (IACUC) protocol (AP#13-MAM1). Embryos were raised and staged according to Kimmel et al.[69]. For injection, 1-cell stage embryos were harvested and injected with mRNAs or MOs in yolk for even

distribution using the typical MPPI-2 quantitative injection equipment (Applied Scientific Instrumentation Co.). In this study, *sec14l3*, *ca-AKT*, and *ca-MEK* mRNA were synthesized using the mMESSAGE mMACHINE Kit (Ambion AM1344) and purified using the RNeasy Mini Kit (Qiagen). Morpholinos were ordered from Gene Tools, LLC. and prepared in stocks. sec14l3-tMO2, sec14l3-sMO, ptp1b-tMO, ptp1b-sMO and std-MO were synthesized by Gene Tools, LLC. The sequences of the used morpholinos are as follows: sec14l3-tMO2: 5′-ATGTCGC-CACGAGTGCAGCAGAAAT-3′; sec14l3-sMO: 5′-ATGTTTCTCACCTCT-CAGCCATCTG-3′; ptp1b-tMO: 5′-CGATTTCCCGAAACTCGGCTTCCAT-3′;

**Fig. 8** Sec14l3/SEC14L2 promote RAB5A/4A activation. **a** Sec14l3 interacts preferentially with GDP-bound state of RAB4A/5 A. HEK293T cells were transfected with Sec14l3-GFP and lysed for in vitro incubation with GTPγS/GDP loaded-RAB4A/5 A respectively. Quantification data of the relative protein levels of bound Sec14l3 are shown on the right ($n = 3$). *$p < 0.05$. **b** Recombinant Sec14l3 protein purified from *E. coli* binds preferentially to GDP-bound RAB4A/5 A in vitro. Recombinant GST-Sec14l3 protein purified from *E. coli* was loaded with GTPγS and immobilized for incubation with the same amount of GTPγS/GDP loaded-RAB4A/5A. Quantification data of the relative protein levels of bound RAB4A/5A are shown on the right ($n = 3$). *$p < 0.05$. **c** Knockdown of *SEC14L2* alleviates RAB5A-GTP levels in HEK293T cells. NC or *SEC14L2* shRNA transfected cell lysates were immunoprecipitated by an anti-RAB5A-GTP antibody or subjected to GST-R5BD pull down. Quantification of the relative protein level of RAB5A-GTP is shown on the right ($n = 3$). **$p < 0.01$, *$p < 0.05$. **d** Knockdown of *SEC14L2* alleviates RAB4A-GTP level in HEK293T cells. NC- or *SEC14L2* shRNA transfected HEK293T cell lysates were immunoprecipitated by anti-RAB4A-GTP antibody and quantification of the relative protein level of RAB4A-GTP is shown on the right ($n = 3$). *$p < 0.05$. *t*-tests were used for statistical analyses in **a**–**d**. **e** Proposed working model of Sec14l3/SEC14L2 for RAB5A/4A-mediated VEGFR2 trafficking. Upon ligand stimulation, VEGFR2 could be internalized and recycled through RAB5A and RAB4A mediated pathways. Sec14l3/SEC14L2 are beneficial for increasing RAB5A/4A in their GTP-bound forms to accelerate the movement of VEGFR2-containing vesicles to early endosomes and subsequently recycling endosomes, consequently preventing VEGFR2 from PTP1B-mediated dephosphorylation

ptp1b-sMO: 5′-GAAGCAGAATCAGTCTTTACCTGGT-3′ and std-MO: 5′-CCTCTTACCTCAGTTACAATTTATA-3′. Every single embryo was injected with 1 nl mRNA or MO solutions at indicated doses and raised to 19–30 hpf for observation. Embryos were harvested randomly for each experiment. For living imaging of vasculature development, chorion was removed from embryos by Pronase K (Sigma, 10 mg ml$^{-1}$) and embryos were embedded in low melting agarose (Amresco 0815). Images were acquired using a Nikon A1RMPSi lasers scanning confocal microscope and processed by NIS-Element software for the measurement of diameters of vessels.

**Whole-mount in situ hybridization and immunofluorescence**. Embryos at indicated stages were harvested and fixed by 4% paraformaldehyde for at least 12 h. After dehydration in methanol gradients, embryos could be stored at −20 degrees for future use. For WISH, at the 1st day, embryos were subjected for rehydration in 0.1% PBST buffer, pre-hybridization in HYB$^-$ buffer and hybridization with the indicated probes overnight at 65degrees; At the 2nd day, embryos were washed by 50% formamide/ 2x SSCT, 2x SSCT, 0.2x SSCT buffer sequentially. After blocking, embryos were incubated with Digoxigenin-AP antibody (11093274910, Roche, 1:3000) overnight at 4 degrees; At the 3rd day, after several rounds of washing by MABT buffer and staining buffer, embryos were transferred into 48-well plate and the substrate BM-Purple (11442074001, Roche) was added for probe visualization. Digoxigenin-UTP−labeled *sec14l3* antisense RNA probe was transcribed from a linearized plasmid in vitro. Stained whole embryos were photographed directly or sectioned in the transverse direction first (Leica CM1900) and then imaged by the Ds-Ri1 CCD camera under a Nikon SMZ1500 stereoscope. For immunofluorescence, mouse anti-GFP (sc-9996, Santa Cruz, 1:100), rabbit anti-p-Erk (Thr202/Tyr204) (#9101, Cell Signaling Technology, 1:100), rabbit anti-p-Akt (Ser473) (#4060, Cell Signaling Technology, 1:100) antibodies were used and embryos were imaged under Nikon A1RMPSi lasers scanning confocal microscope.

**Cell culture and transfection**. Plasmids used for cell transfection and their primers for construction are listed in Supplementary Table 1. HEK293T cells (Cell Resource Center, Peking Union Medical College) were cultured in DMEM (Life Technologies) supplemented with 10% FBS (Hyclone, Logan, UT) and 50 mg ml$^{-1}$ penicillin/streptomycin (PS) (Invitrogen). HUVECs (ScienCell #8000) and HUAECs (ScienCell #8010) were cultured in endothelial cell medium (ScienCell #1001). Cell lines were checked for free of mycoplasma contamination by PCR and culture. Its species origin was confirmed with PCR. The identity of the cell line was authenticated with STR profiling (FBI, CODIS). The results for HEK293T cells can be viewed on the website (http://cellresource.cn). Transfections were performed using the polyethylenimine method for HEK293T cells. For endothelial cells, lentivirus infection method was used for knockdown experiments.

**In vitro tube formation and wound closure assay**. For in vitro tube formation assay, NC-shRNA or SEC14L2-shRNA infected HUVEC cells were plated on 12-well plate ($2 \times 10^5$ cells per well) covered by Matrigel (BD 356230) and cultured for assessment of tube formation following seeding. The tube number was counted. For wound closure assay, HUVEC cells were seeded in 6-well plate and grown to confluent state. Then, a 'scratch' was created using the P200 pipet tip and the closure of the wound was monitored after 24 or 48 h to reflect the migration of endothelial cells.

**Internalization and recycling assay**. To quantify internalization of VEGFR2, biotin based biochemical quantification and anti-VEGFR2 antibody-based immunofluorescence were adopted. The biotin based assay was performed essentially as described[12]. Generally, biotin in the culture medium can bind to proteins including VEGFR2 on the cell surface at 4 °C, but the biotin-labeled proteins would not be internalized at this temperature. After free biotin molecules are washed away and incubation temperature shifts to 37 °C, the biotin-labeled proteins on the cell surface start to move into the cytoplasm through internalization. The intracellular/internalized biotin-labeled proteins can be pulled down using streptavidin beads

after removal of those retained on the cell surface with iodoacetamide treatment and specific protein associated with biotin can be detected using a specific antibody at different time points of internalization (Supplementary Fig. 10). Briefly, transfected cells were washed twice using cold PBS (pH 8.0) and surface proteins were covalently labeled using a membrane-impermeant biotinylation reagent NHS-SS-biotin (Pierce, Rockford, IL) at 4 °C for 30 min. After labeling, un-reacted biotin was washed away by chilled PBS and labeled cells were immediately incubated at 37 °C to allow internalization for 5 min or 10 min. After internalization, culture media was discarded and washed sequentially by 10% FBS twice, 1% BSA once and 20 mM iodoacetamide (IAA) once for 10 min to remove biotin-labeled proteins remaining on the plasma membrane. Then, cells were lysed for streptavidin beads pull-down assay. For VEGFR2 immunofluorescence assay, HUVECs infected with shRNA were cultured for 72 h and then washed twice by cold PBS, which was followed by incubation with the Alexa647-conjugated anti-VEGFR2 antibody (#359909 Biolegend, 1:200) for 30 min at 4 °C. After removing extra antibody, cells were placed in an incubator at 37 °C for internalization. At indicated time points, cells were washed with acid PBS (pH 2.5) for three times to remove un-internalized anti-VEGFR2 antibody on the cell surface. After wash, cells were fixed by 4% PF and subjected for immunofluorescence. Antibodies used here including SEC14L2 (TA503723, ORIGENE, 1:100), EEA1 (#3288, Cell Signaling Technology, 1:100), VEGFR2 (#2479, Cell Signaling Technology, 1:200) and CD63 (#CBL553, Millipore, 1:100).

To measure the recycling of VEGF2, transfected cells were washed and labeled at 4°C with biotin for 30 min. The labeled cells were incubated with pre-warmed culture medium for 20 min at 22°C to allow cell surface proteins to internalize into RAB4-positive vesicles. The biotin-labeled proteins retained on the cell surface were removed by sequential washes with 10% FBS twice, 1% BSA once and 20 mM iodoacetamide (IAA) once, each for 10 min at 4 °C. Then cells were incubated at 37 °C to allow recycling in the absence or presence of 100 ng ml$^{-1}$ VEGFa for 15 min. After recycling, cells were placed on ice and subjected to wash with 10% FBS twice, 1% BSA once and 20 mM iodoacetamide (IAA) once, each for 10 min, to remove labeled proteins that are returned to the plasma membrane. Finally, cells were lysed by TNE buffer for streptavidin beads pull-down assay and samples were boiled with 2x loading buffer and prepared for western blotting.

**Co-immunoprecipitation and GST pull-down**. Co-IP was carried out according to the previous paper[70]. Briefly, cells were lysed using TNE buffer (150 mM NaCl, 5 mM EDTA, 1% TritonX-100, 10 mM Tris-HCl [pH 7.4]) mixed with protease inhibitors. The cell lysate was centrifuged at $120000 \times g$ to remove cell debris to acquire supernatant for subsequent SDS-PAGE or IP assay. For the co-IP assay, the following antibodies were used: anti-VEGFR2 (#2479, Cell Signaling Technology, 1:200), anti-RAB4 (ab109009, Abcam, 1:100), anti-RAB5 (ab18211, Abcam, 1:100), anti-Flag (F1804, Sigma, 1:1000 for western blot). For immunoblotting analysis, anti-p-VEGFR2-Y$^{1175}$ (#2478, Cell Signaling Technology, 1:2000 for western blot), anti-p-VEGFR2-Y$^{1054/1059}$ (ab5473, Abcam, 1:1000 for western blot), anti-GFP (sc-9996, Santa Cruz, 1:1000 for western blot), anti-SEC14L2 (TA503723, ORIGENE, 1:1000 for western blot) and anti-PTP1B (610139, BD Bioscience, 1:1000 for western blot). For GST pull-down assay, GST-Sec14l3 and GST-RAB5A/4A recombinant proteins were expressed in *E. coli*. After 16 h induction by 1 mM isopropyl β-D-1-Thiogalactopyranoside (IPTG, Takara 9030) at 16 °C, bacterial culture was centrifuged at $5000 \times g$ and washed by 50 mM Tris-HCl (pH 7.4) twice and then sonicated in 50 mM Tris-HCl (pH 7.4) solution supplemented with 1 mM PMSF. Next, the supernatant was incubated with glutathione-Sepharose beads (GE Healthcare), washed by 50 mM Tris-HCl (pH 7.4), 50 mM Tris-HCl (pH 7.4) with 500 mM NaCl sequentially. After wash, about 1 ml PreScission Protease (GE Healthcare) was added into GST-RAB5A/4 A beads to cut the GST tag and the resulting RAB5A/4 A protein was eluted using 50 mM Tris-HCl (pH 7.4) solution. For GTPγS/GDP- RAB5A/4 A pull-down assay, RAB5A/4 A protein was incubated in 20 mM HEPES (pH 7.5), 100 mM NaCl, 10 mM EDTA, 5 mM MgCl$_2$, 1 mM DTT, and 1 mM GTPγS/GDP (freshly made), for 90 min at room temperature to load GTPγS/GDP. Then GTPγS incubated GST-Sec14l3 beads were added into the solution for another 2 h incubation at 4 °C. Beads were finally washed and

harvested for 2x loading buffer boiling and the samples were used for SDS-PAGE and Coomassie blue staining.

**Statistical analysis**. Results were analyzed with GraphPad Prism 5.0 and are presented as mean ± SEM/SD as indicated, and comparisons were performed between two groups using a two-tailed Student's $t$-test or ANOVA test when comparing more than two conditions. For all analyses, $*p < 0.05$; $**p < 0.01$; $***p < 0.001$ were considered statistically significant; ns indicated statistical non-significance with $p > 0.05$. Each experiment was carried out at least three times independently. The sample size is pre-estimated to ensure statistical analysis and no sample was optionally excluded from analysis. No blinding was done in the analyses and quantifications.

## Data availability

The authors declare that all data supporting the findings of this study are available within the article and its supplementary information files or from the corresponding author upon reasonable request. RNA-Seq raw data have been deposited in the GEO database under accession code: GSE126617 and other raw images are available at figshare.com for: Fig. 1d, https://doi.org/10.6084/m9.figshare.7729319, Fig. 2b, https://doi.org/10.6084/m9.figshare.7729325, Fig. 2g, https://doi.org/10.6084/m9.figshare.7729331, Fig. 3d, https://doi.org/10.6084/m9.figshare.7729334, Fig. 3e, https://doi.org/10.6084/m9.figshare.7729337, Fig. 5e, https://doi.org/10.6084/m9.figshare.7729340. The source data underlying Figs. 2, 3, 4, 5, 6, 7 and Supplementary Fig. 4 are provided as a Source Data file.

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

## Acknowledgements

We thank Drs. Nathan D. Lawson (University of Massachusetts Medical School) and Bo Zhang (Peking University) for providing transgenic fishes and Wang Min (Yale University) for providing WT-VEGFR2 and KM-VEGFR2 plasmids. We also appreciate the helpful discussion and technical assistance from members of the Meng Laboratory. This work was financially supported by grants from the National Natural Science Foundation of China (#31522035, #91754112, #31801130 and #31371460), Major Science Programs of China (#2012CB945100).

## Author contributions

B.G. performed most experiments, collected and organized data. Z.H.L. helped to culture cells for partial biochemical analysis. W.H.X., G.Y.L., and S.H.D. participated in a part of the work, including whole-mount in situ hybridization, constructs cloning and so on. A.M.M. and S.J.J. proposed and managed the project, and B.G., A.M.M., and S.J.J. designed the study and prepared the manuscript.

## Additional information

**Competing interests:** The authors declare no competing interests.

