## [Peer Review File · Nature Communications]

Reviewers' comments:

Reviewer #1 (Remarks to the Author):

The authors identified sec14I3/sec14I2 as a gene highly expressed in zebrafish ECs during vessel growth and in cultured HUVECs. To examine the role of sec14I3 in endothelial cells, the authors knockdown the expression of this protein with morpholino oligonucleotides induction as well as the cas9/crispr gene targeting strategy and confirmed the sprouting behavior of ECs was compromised. Interestingly, vascular defects in DA were restored by MEK overexpression, while compromised endothelial behavior at PCV was partially recovered with overexpression of AKT but not MEK. The authors identified that in sec14I3 loss of function condition, VEGFR2 autophosphorylation at Y1054/1059 was not affected but that at Y1175 was downregulated. Furthermore, they identified sec14I3 as a novel binding partner of Rab5A/4A and as a regulator of VEGFR2 trafficking.

The overall experiments were carefully designed and done and the manuscript reports a number of interesting results. Unfortunately the current manuscript does not contain a great conceptual advancement. In the last 10 years, many factors have been shown to control endothelial sprouting behavior in the mouse retina and zebrafish DA and PCV. The authors added one more factor in this series of studies. Related this, the specificity of the upstream signaling of sec14I3 is not considered in this manuscript. Moreover, the specific effect on VEGFR2 Y1175 but not on Y1054/1059 is not addressed.

Major comments

1. The biochemical analysis using HUVECs are generally well done. However, I could see two big gaps. Chiefly, specifically defected Y1175 phosphorylation is not explained by any data presented. Secondly, it is unclear to which extent the in vivo observations could be explained by defected VEGFR2 trafficking. For instance, VEGFR3 is also important endothelial behavior. It often forms a complex with VEGFR2. Other growth factor receptor such as FGF receptors might also play roles.
2. The authors have shown the different role of MEK and AKT signaling in DA and PCV respectively. AKT and MEK signaling are controlled by not only growth factor signaling, for instance, but also shear stress. Other factors are not considered.
3. HUVECs are primary cultured ECs derived from human umbilical vein. The authors presented different characteristics of ECs in DA and PCV, which is very interesting and new. On the other hand, the in vitro experiment was done by only using HUVEC

Minor comments

1. Most of statistics analysis in the manuscript must be done by the ANOVA test but not t-test.
2. In Figure 5b, total number of endosome seems to be decreased after shRNA KD. This should be quantified or the panels should be replaced.

Reviewer #2 (Remarks to the Author):

The major claim of this paper is that a lipid transporter, Sec14I3/SEC14L2 is essential for the formation of the vasculature during embryogenesis in zebrafish. Formation of the vasculature is dependent on VEGF receptor 2 signalling. Their results suggest that SEC14L2 acts as a Guanine nucleotide exchange factor (GEF) for two RAB GTPases, RAB4 and RAB5 and thus modulate the rapid internalisation of the activated receptor from the plasma membrane followed by fast recycling of the VEGFR2 from the RAB5 endosomal compartment to the RAB4 compartment for recycling the receptor back to the plasma membrane. In the absence of this rapid recycling, the VEGFR2 receptor signalling is defective and affects lumen formation of both the dorsal aorta and the posterior cardinal vein.

Much is known about VEGFR2 signalling and the regulation of vasculogenesis including the importance of the phospholipase C/ERK signalling as well as PI3K/Akt signalling and the

requirement for the fast recycling of the VEGFR2 receptor. Delay in the fast recycling of the receptor leads to the dephosphorylation by PTB phosphatase and inactivation of the receptor and thus leads to defects in vasculogenesis. What is novel in this paper is the identification of a new regulator, which is a lipid transporter that has a hydrophobic cavity for binding lipids including alpha-tocopherol (vitamin E) and phosphatidylinositol. SEC14L2 (also known as TAP or SPF) is known to bind tocopherol and this binding can be completed by phosphatidylinositol. Moreover, SEC14L2 has been shown to have GTPase activity in previous studies by Zinng and Azzizi et al. A recent study by Gong et al (Elife 2017) confirmed that Sec14L2 binds GTP and possess GTPase activity and moreover, activates phospholipase Cdelta4 to initiate Ca²⁺ signalling. In that study they also showed that SEC14L forms a complex with Frizzled and Dishevelled, proteins involved in the Wnt signalling pathway.

In this new study, they show that SEC14L2 can physically interact with the VEGF2 receptor and bind to both RAB4 and RAB5 and act as a GEF (Guanine nucleotide exchange Factor) for these proteins. Thus SEC14L2 is not only a GTPase (hydrolyses GTP) (shown in their previous study) but also a GEF (exchange factor for RAB4 and 5) shown in this new paper. This makes SEC14L2 a versatile protein whereby it not only acts as transducer of signalling by activating PLC-delta but also an exchange factor for RABS coupled with the fact it also binds tocopherols and phosphatidylinositol.

What the paper shows:

Fig. 1 - SEC14L2 is enriched in endothelial cells and in the vasculature of zebrafish. (Expression restricted to PCVs (posterior cardinal vein).

Fig. 2 - two morpholinos (used to reduce sec14l3 in fish -tMO2 and sMO) reduces a sub-population of angioblasts to be sorted and segregated to form the arterial and venous compartments. The DA and PCV compartments undergo rapid luminal expansion and this is compromised in Sec14L3 morpholino injected cells. Morpholino effects rescued by injection of mRNA. Migration and tube formation is affected in HUVECS after SEC14L2 knockdown.

Figure 3 - Knockdown of SEC14L2 inhibits VEGF stimulated ERK and AKT phosphorylation in HUVECs. pERK is reduced in morpholino-treated fish. Dorsal diameter rescued by constitutively active MEK and PCV rescued by constitutively active Akt in fish.

Figure 4 - VEGF phosphorylated Tyr1175 and inhibited by SEC14L2 k/d and rescued by over expression of SEC14L2 or by down regulation of PTP-1B phosphatase in HUVECS.

Figure 5 - Internalisation of the VEGF receptor is slower in knockdown HUVEC cells. Receptor internalisation and recycling rates are slowed down in knockdown cells.

Fig 6 - SEC14L2 via its SEC14 domain physically interacts with the activated VEGFR and Sec14L3 binds to Rab4A and Rab5A - no particular domain required

Fig 7 - SEC14L3 prefers to bind RABs in the GDP form and activates loading of RABs.

Comments

The story as presented is backed up by solid experimental data and makes a convincing argument about the function of Sec14l3/SEC14L2 in vasculogenesis. Some points of detail need to be addressed.

1. Fig. 1d - the localisation of SEC14L2 with VEGFR2 is not convincing. The SEC14L2 appears to be present throughout the cell including the nucleus with no specific localisation whilst the majority of the VEGFR2 appears to be localised to the Golgi.

2. If the function of Sec14L2 was to internalise and recycle the VEGF receptor to the plasma membrane, it would be expected that in the absence of Sec14L2, there would be less VEGFR2 as it would be degraded in the lysosomes? (Its trafficking to the lysosomes would be increased.) Does knockdown of SEC14L2 lead to a decrease in VEGFR2 levels? Do they find that the VEGFR2 receptor is now present in Rab7 containing endosomes?

3. In Figure 6f, a truncated RAB5 is used for immunoprecipitation. What was the rationale for using a truncated form? Results with a full length protein should be provided.

4. The authors have to discuss the finding from their previous paper (Gong et al Elife 2017) and to

make sense of how SEC14L2 can be both a GTP binding protein transducing signals to activate downstream effectors and also act as an exchange factor for RAB proteins.

5. It would be appropriate to reference some of the early work on SEC14L2 (also known as TAP1 and SPF), in particularly since tocopherols are thought to influence both angiogenesis and vasculogenesis.

6. Can the two activities, GTPase activity and the GEF activity of the SEC14L2 proteins be better defined ? Can the two activities be separated out (i.e different domains required or specific amino acids).

Minor Comments:

Line 128, 701, 708 - CARL-TRIO domain should be CRAL-TRIO domain throughout the manuscript

Line 617 - 'Filed, should be 'field'

Line 161 and line 275 - PLCgamma/MAPK/ERK. ERK is MAPK.

Line 166 - consistent with the reduction of activated p42/p44 MAPK and PI3K/AKT. They do not show reduction in PI3K but only reduction in phosphorylated Akt.

Line 353 - Reference for tube formation assay in Huvecs

Line 370 - spelling of Alexa647

Fig. 6a and 6c- molecular weight markers of Sec14L2 and VEGFR appear to be mixed up in Total Cell Lysates blots

Figure 6J - the total cell lysates and the IP Flag pull down appear to be mislabelled ?

Reviewer #4 (Remarks to the Author):

The manuscript by Gong et al. suggests a role for SEC14L2/Sec14I3 in the trafficking of VEGFR2 and thus in vascular development. The proposed mechanism involves direct binding of SEC14L2 to VEGFR2 on the one hand and RAB4A/5A on the other hand. The latter binding increases RAB4A/5A GTPase activity and internalization and recycling of VEGFR2. These observations are confirmed by mostly convincing biochemical data. Furthermore, the authors suggest a role of Sec14I3 in vasculogenesis and angiogenesis in zebrafish embryos, as Sec14I3 morphants show reduced vessel size and display defects in DA and PCV morphogenesis. This *in vivo* part is less convincing, since the phenotypes of the mutants appear relatively mild and are not always consistent with the data the authors present when using morpholino-mediated knockdown of sec14I3. Overall, the manuscript is well written and easy to follow and most of the data is presented in a clear way. It would still benefit from a careful proofreading. There are several issues with experimental interpretation, image quantification and statistical analysis, which the authors should address.

1. The authors analyze the formation of the first trunk blood vessels, the DA and the PCV (Figure 2a). Here, they observe an impairment of sprouting of the PCV from the DA in sec14I3 MO injected embryos. These data need to be quantified. Do mutants show the same phenotype? In addition, a report put the "ventral sprouting" concept into question by showing that DA and PCV precursors originate in different locations in the early embryo before migrating to the embryonic midline (Kohli et al., Arterial and venous progenitors of the major axial vessels originate at distinct locations, *Dev. Cell* (2013)). Do the authors observe specific defects in early angioblast populations prior to their migration to the midline (e.g. at the 10-18 somite stage)?

2. There are some issues with the analysis of the vascular phenotypes in the trunk of zebrafish embryos. The authors show that morpholino injected embryos show shortening of the DA, PCV and ISVs (Supplementary Figure 2). However, for the sec14I3 mutants, they only analyze the length of the DA and PCV and not of the ISVs. In addition, they do not present the data as actual values

(how many μm), but only as a “relative” length. The authors need to provide the data for their length measurements. Is there a difference in ISV length between morpholino injected embryos and mutants? In addition, to verify the morpholino phenotype, the authors need to inject the morpholino into *sec14l3* mutants and measure ISV length. How do the values compare to the morpholino and mutant values?

3. The authors also only show data for the lumen diameters in morpholino injected embryos, not in the mutants. The analysis for lumen diameters also needs to be repeated in mutants. From Figure 2d, it seems that ISV sprouting is only marginally affected (see point 2). Can the authors provide an explanation, why a process that heavily relies on VEGF signaling is only very mildly affected in their mutants? Can the authors analyze maternal/zygotic mutants, which might show a more severe phenotype in vascular development? In addition, a previous report showed that ERK phosphorylation affects endothelial migration in zebrafish ISVs (Shin et al., 2016, Development “Vegfa signals through ERK to promote angiogenesis, but not artery differentiation”). The authors convincingly show reduction in ERK phosphorylation. Why is ISV migration not affected?

4. In HUVECs, both ERK and AKT phosphorylation are affected by *sec14l3* knockdown (Figure 3). For zebrafish, the authors only analyze ERK phosphorylation, but not AKT phosphorylation. Is AKT phosphorylation affected in mutant/morpholino injected zebrafish embryos?

5. The authors do not show any pictures depicting the vasculature of *ca*-AKT and *ca*-MEK injected embryos. How about overall body shape and blood flow in these embryos? Also, the description in the figure legend is not clear. Does the figure include the values from 3 independent experiments or just one experiment? Did the other two experiments show the same results? What values were used for statistical analysis? Are these from different fish in one experiment or different vessel regions of the same fish? Differential effects of *ca*-AKT and *ca*-MEK in different vessels is potentially an interesting finding. However, the authors do not address it further, neither experimentally, nor in the discussion.

6. The authors claim that *sec14l3* limits the exposure of phosphorylated VEGFR2 to phosphatases, such as PTP1B. To distinguish between phosphatase exposure and a possible role of *sec14l3* in protein degradation, the authors need to analyze the effect of inhibiting protein degradation on VEGFR2 phosphorylation after knockdown of *sec14l3* (e.g. via inhibiting the proteasome).

7. The authors use a complex biotin labelling assay to analyse VEGFR2 recycling (Fig. 5d). The lysates IN2 show a reduced amount of biotinylated VEGFR2 in NC-shRNA infected cells compared to IN1 lysate. This observation is attributed to VEGFR2 recycling to the cell membrane. However, can the authors exclude the degradation of VEGFR2 upon internalization since they write in the discussion that VEGFR2 can undergo lysosomal degradation? Is it possible to observe the same changes in intracellular VEGFR2 upon inhibition of protein degradation? Also, is it possible to measure biotinylated VEGFR2 only on the plasma membrane (before and after internalization)? The interpretation of the assay is important for understanding the role of SEC14L2, which shows a clear effect in this experiment. Could you also explain why there seems to be more biotinylated VEGFR2 upon internalization (IN1) than before (PM).

8. The authors claim that *sec14l3* preferentially binds to GDP bound RAB4A and RAB5A performing binding assays in the presence of GTP γ S. However, in a previous report (Gong et al., 2017), the authors showed that *sec14l3* itself shows GTPase activity. How can they exclude that the GTP γ S inhibits *sec14l3* GTPase activity and thereby diminishes interaction with RAB4A and RAB5A?

Minor:

1. The authors motivate their study with an RNA-Seq experiment, however they do not show any

results. It is also not clear, whether the expression of sec14l3 was detected using RNA-Seq. The results should be either shown or the RNA-Seq should not be mentioned.

2. In the abstract the authors state "However, little is known about how Sec14-like phosphatidylinositol transfer proteins (PITPs) are involved in this process." It is not obvious why PITPs would be involved in VEGF signalling. An additional sentence about the study rationale could make it clearer.

3. The description of the biotinylation assay should be moved from the results section to the materials and methods section.

4. The sentence in the Discussion "Of these known regulators, either VEGFR2 internalization or degradation is affected" could be rephrased to improve the clarity.

Reviewer #1 (Remarks to the Author):

The authors identified sec14l3/sec14l2 as a gene highly expressed in zebrafish ECs during vessel growth and in cultured HUVECs. To examine the role of sec14l3 in endothelial cells, the authors knockdown the expression of this protein with morpholino oligonucleotides induction as well as the cas9/crispr gene targeting strategy and confirmed the sprouting behavior of ECs was compromised. Interestingly, vascular defects in DA were restored by MEK overexpression, while compromised endothelial behavior at PCV was partially recovered with overexpression of AKT but not MEK. The authors identified that in sec14l3 loss of function condition, VEGFR2 autophosphorylation at Y1054/1059 was not affected but that at Y1175 was downregulated. Furthermore, they identified sec14l3 as a novel binding partner of Rab5A/4A and as a regulator of VEGFR2 trafficking.

The overall experiments were carefully designed and done and the manuscript reports a number of interesting results. Unfortunately the current manuscript does not contain a great conceptual advancement. In the last 10 years, many factors have been shown to control endothelial sprouting behavior in the mouse retina and zebrafish DA and PCV. The authors added one more factor in this series of studies. Related this, the specificity of the upstream signaling of sec14l3 is not considered in this manuscript. Moreover, the specific effect on VEGFR2 Y1175 but not on Y1054/1059 is not addressed.

Major comments

1. The biochemical analysis using HUVECs are generally well done. However, I could see two big gaps. Chiefly, specifically defected Y1175 phosphorylation is not explained by any data presented. Secondly, it is unclear to which extent the *in vivo* observations could be explained by defected VEGFR2 trafficking. For instance, VEGFR3 is also important endothelial behavior. It often forms a complex with VEGFR2. Other growth factor receptor such as FGF receptors might also play roles.

Answer: We thank the reviewer very much for such valuable advice.

1) So far, many phosphatases have been discovered to be able to extinguish VEGFR2 phosphorylation at its tyrosine residues in cytoplasmic domain^{1, 2, 3, 4, 5, 6}. In particular, PTP1B has been reported to specifically dephosphorylate the Y1175 site⁵. We have shown that PTP1B co-knockdown could improve defected Y1175 phosphorylation in SEC14L2 knockdown cells *in vitro* (Fig. 4g, h) and rescue lumen size of DA and PCV in sec14l3 morphants *in vivo* (Fig. 4k, l), indicating that the specifically defected VEGFR2 Y1175 phosphorylation attributed to its key regulator PTP1B. To directly investigate whether decreased phosphorylation of VEGFR2 Y1175 was caused by prolonged exposure of the activated VEGFR2 to PTP1B, we performed co-IP experiments to compare their binding affinities at different expression levels of SEC14L2 in HEK293T cells. The results showed that SEC14L2/Sec14l3 could indeed prevent VEGFR2 from interacting with PTP1B (Fig. 4f), therefore protecting VEGFR2 Y1175 from de-phosphorylation. We have included these new data in the revised manuscript.

2) VEGFR1/2/3 play critical roles in vasculature development and there are various cross-talks among them to achieve accurate signal outputs. Although we could not exclude all other alternatives, we found that Sec14l3 interacts with the cytoplasmic region of VEGFR2, but not those of VEGFR1 and VEGFR3 (Fig 6a), indicating that these *in vivo* observations should be somehow specifically dependent on VEGFR2, but not VEGFR1 and VEGFR3 signaling. What's more, VEGFR2-cyto mRNA injection also could restore luminal phenotypes in sec14l3 morphants (Fig. 4i, j). Therefore, we conclude that these phenotypes in sec14l3 deficient embryos should be largely due to the defected VEGFR2 signaling. We have included these new data in the revised manuscript.

3) As the reviewer mentioned that blood vessels formation is a multistep process dependent on a number of growth factors, including VEGF, FGF and so on. FGF signaling, acting through cell-surface tyrosine kinase receptors, has been implicated in several processes of vascular system formation, including the angioblasts induction from the mesoderm, the process of angiogenesis and the maintenance of vascular integrity^{7, 8, 9}. However, *etsrp* or *fli* labelled angioblasts are specified in the mesodermal germ cell layer normally in *sec14l3* mutants (Supplementary Fig 5). Additionally, ISV sprouting and vascular integrity are also not obviously affected in the mutants (Fig 2d, Supplementary Fig 4, and data not shown). All of these results suggest that FGF signaling may be not significantly affected in our case.

2. The authors have shown the different role of MEK and AKT signaling in DA and PCV respectively. AKT and MEK signaling are controlled by not only growth factor signaling, for instance, but also shear stress. Other factors are not considered.

Answer: Thanks for the reviewer's suggestion. In this manuscript, relationships between VEGFR2-MEK/AKT and Sec14l3/SEC14L2 are mainly explored. Our new rescue data of *VEGFR2-cyto* mRNA (Fig. 4i, j) indicates that MEK and AKT signaling responsible for luminal defects in *sec14l3* mutants should be largely derived from VEGFR2 transduction. We have already added these new data in our revised manuscript. Other factors, either growth factor or shear stress, are not fully considered in our study, although both of them can act through VEGF-VEGFR2 axis to control AKT and MEK branches¹⁰. It will be of great interest to study the functions of *sec14l3* in regulating other growth factors or shear stress pathways independent of VEGFR2 signaling in the future.

3. HUVECs are primary cultured ECs derived from human umbilical vein. The authors presented different characteristics of ECs in DA and PCV, which is very interesting and new. On the other hand, the in vitro experiment was done by only using HUVEC.

Answer: We thank the reviewer for this advice and investigated *SEC14L2* knockdown effects in human umbilical arterial endothelial cells (HUAECs). As shown in Supplementary Figure 7, *SEC14L2* knockdown mainly counteracts VEGFa-motivated p-AKT level, whereas the p-ERK level was not changed a lot in HUAECs (Supplementary Fig. 7). We have included these new data in our revised manuscript.

Minor comments

1. Most of statistics analysis in the manuscript must be done by the ANOVA test but not t-test.

Answer: We thank the reviewer for this criticism, and ANOVA test was used where appropriate in our new version.

2. In Figure 5b, total number of endosome seems to be decreased after shRNA KD. This should be quantified or the panels should be replaced.

Answer: We thank the reviewer for this suggestion and have added the quantification data (Fig. 5c).

Reviewer #2 (Remarks to the Author):

Comments

The story as presented is backed up by solid experimental data and makes a convincing argument about the function of Sec14l3/SEC14L2 in vasculogenesis. Some points of detail need to be addressed.

1. Fig. 1d – the localisation of SEC14L2 with VEGFR2 is not convincing. The SEC14L2 appears to be present throughout the cell including the nucleus with no specific localisation whilst the majority of the VEGFR2 appears to be localised to the Golgi.

Answer: Previous studies have revealed that the internal pool of VEGFR2 corresponds to at least 3 distinct compartments, the largest being EEA1⁺/Rab4⁺ sorting/recycling endosomes, a smaller but significant pool of CD63⁺ late endosomes and Golgi^{11,12}. Although there is no obvious co-localization in the Golgi apparatus, about 28% of SEC14L2 shows colocalization with VEGFR2 in the endocytic pool. We thank the reviewer very much for this comment and have rewritten this part more accurately in our revised manuscript.

2. If the function of Sec14L2 was to internalise and recycle the VEGF receptor to the plasma membrane, it would be expected that in the absence of Sec14L2, there would be less VEGFR2 as it would be degraded in the lysosomes? (Its trafficking to the lysosomes would be increased.) Does knockdown of SEC14L2 lead to a decrease in VEGFR2 levels? Do they find that the VEGFR2 receptor is now present in Rab7 containing endosomes?

Answer: This reviewer raised a very good question and gave very constructive suggestion. In order to investigate whether the internalized VEGFR2 could be present in late endosomes or not, we performed a co-localization assay of VEGFR2 with CD63 or RAB7, widely defined markers of late endosomes in HUVECs, in the absence or presence of SEC14L2. No matter which marker was used, we found that knockdown of SEC14L2 led to VEGFR2 accumulation in late endosomes (Fig. 5e, f). Based on our new western blotting results showing that knockdown of SEC14L2 has no significant effect on the VEGFR2 total protein level, it is indicated that VEGFR2 accumulates in late endosomes without further degradation (Supplementary Fig. 11). This notion is consistent with the previous report¹¹, which proposes that CD63⁺/VEGFR2⁺ endosomes can act as a storage compartment for VEGFR2, without necessarily delivering the receptor to the lysosome for degradation. We have added these new results in our revised manuscript.

3. In Figure 6f, a truncated RAB5 is used for immunoprecipitation. What was the rationale for using a truncated form? Results with a full length protein should be provided.

Answer: We apologize for this inconsistency. When performing the immunoprecipitation assay in Figure 6f, the full-length RAB5 is not available for us, so we used the truncated form instead. According to the reviewer's suggestion, co-IP experiments of SEC14L2 with full-length RAB proteins were carried out and the new result was provided in Figure 6f. Besides, Fig 6h, 7a, 7b also show the interactions between SEC14L2 and full-length RAB5.

4. The authors have to discuss the finding from their previous paper (Gong et al Elife 2017) and to make sense of how SEC14L2 can be both a GTP binding protein transducing signals to activate downstream effectors and also act as an exchange factor for RAB proteins.

Answer: We thank the reviewer for this question. To prove whether SEC14L2/Sec14l3 can act as a GEF for RAB5 protein, we expressed and purified recombinant Sec14l3 and RAB5 protein from *E.coli* to perform GTP exchange assay based on mant-GTPγS fluorescence^{13, 14, 15, 16}. Raw data and fitted one-phase exponential association curves are shown in Supplementary Figure 12. Consistent with our previous reports¹⁷, following the addition of 0.8 μM Sec14l3 itself, a time-course of fluorescence increase could be observed (green curve), indicating its GTP binding

activity. When measuring fluorescence upon addition of 2 μ M RAB5 protein or 2 μ M RAB5 and 0.8 μ M Sec14I3 protein mixture, there is no significant difference (blue and red curves), suggesting that Sec14I3 is not a GEF for RAB5. However, there are still other possible mechanisms that Sec14I3 could promote GTPase activity of RAB5, including serving as a modulator to enhance interactions between RAB5 and its GEF proteins, such as Rbex5, RABGEF1 and so on^{18, 19, 20, 21, 22, 23}, which needs to be further investigated. We have added this new data and discussed the possibilities in our revised manuscript.

5. It would be appropriate to reference some of the early work on SEC14L2 (also known as TAP1 and SPF), in particular since tocopherols are thought to influence both angiogenesis and vasculogenesis.

Answer: Thanks for the reviewer's reminding and we have added the related references.

6. Can the two activities, GTPase activity and the GEF activity of the SEC14L2 proteins be better defined? Can the two activities be separated out (i.e different domains required or specific amino acids).

Answer: We have demonstrated that Sec14I3/SEC14L2 does not work as a GEF for RAB5 (Please see above, Comment 4). As for the GTPase activity, it was proved that the G α subunit (260 aa - 276 aa) in the SEC14 domain of Sec14I3 responses for its GTPase activity in our previous eLife paper¹⁷.

Minor Comments:

Line 128, 701, 708 - CARL-TRIO domain should be CRAL-TRIO domain throughout the manuscript

Line 617 – 'Filed, should be 'field'

Line 161 and line 275 – PLCgamma/MAPK/ERK. ERK is MAPK.

Line 166 – consistent with the reduction of activated p42/p44 MAPK and PI3K/AKT. They do not show reduction in PI3K but only reduction in phosphorylated Akt.

Line 353 - Reference for tube formation assay in Huvecs

Line 370 – spelling of Alexa647

Fig. 6a and 6c– molecular weight markers of Sec14L2 and VEGFR appear to be mixed up in Total Cell Lysates blots

Figure 6J – the total cell lysates and the IP Flag pull down appear to be mislabelled?

Answer: We thank the reviewer very much and apologize for our mistakes. We have corrected all of them in our revised manuscript.

Reviewer #4 (Remarks to the Author):

The manuscript by Gong et al. suggests a role for SEC14L2/Sec14I3 in the trafficking of VEGFR2 and thus in vascular development. The proposed mechanism involves direct binding of SEC14L2 to VEGFR2 on the one hand and RAB4A/5A on the other hand. The latter binding increases RAB4A/5A GTPase activity and internalization and recycling of VEGFR2. These observations are confirmed by mostly convincing biochemical data. Furthermore, the authors suggest a role of Sec14I3 in vasculogenesis and angiogenesis in zebrafish embryos, as Sec14I3 morphants show reduced vessel size and display defects in DA and PCV morphogenesis. This in vivo part is less convincing, since the phenotypes of the mutants appear relatively mild and are not always consistent with the data the

authors present when using morpholino-mediated knockdown of *sec14l3*. Overall, the manuscript is well written and easy to follow and most of the data is presented in a clear way. It would still benefit from a careful proofreading. There are several issues with experimental interpretation, image quantification and statistical analysis, which the authors should address.

1. The authors analyze the formation of the first trunk blood vessels, the DA and the PCV (Figure 2a). Here, they observe an impairment of sprouting of the PCV from the DA in *sec14l3* MO injected embryos. These data need to be quantified. Do mutants show the same phenotype? In addition, a report put the “ventral sprouting” concept into question by showing that DA and PCV precursors originate in different locations in the early embryo before migrating to the embryonic midline (Kohli et al., Arterial and venous progenitors of the major axial vessels originate at distinct locations, *Dev. Cell* (2013)). Do the authors observe specific defects in early angioblast populations prior to their migration to the midline (e.g. at the 10-18 somite stage)?

Answer: We appreciate the reviewer very much for these constructive suggestions.

1) Quantification data for ventral sprouting defects of venous endothelial progenitor cells in *sec14l3* morphants have been added in Supplementary Figure 6. Additionally, we also observed similar ventral sprouting defects in *sec14l3* mutant embryos (Fig. 2g), and have included these new results in our revised manuscript.

2) During developmental vasculogenesis, two important processes have been reported as the reviewer mentioned. Angioblasts originated in the lateral plate mesoderm (LPM) migrate to the dorsal midline to form the DA and PCV in two waves²⁴. In addition, some of venous endothelial cells in dorsally positioned vascular cord migrate ventrally to the PCV after 21 hpf by the mechanism of ventral sprouting²⁵, which has been demonstrated to be decreased in our *sec14l3* deficient embryos from 21 hpf (Fig. 2g, and Supplementary Figure 6). According to the reviewer’s suggestion, we performed whole-mount in situ hybridization, using *fli* and *etsrp* as probes to visualize angioblasts at the 10-16 somite stages in *sec14l3* morphants and mutants. The results revealed that angioblasts migration but not specification is affected in *sec14l3* deficient embryos (Fig. 2f and Supplementary Fig. 5). What’s more, the angioblasts migration defects could be rescued by the injection of *sec14l3* mRNA (Fig. 2g). Therefore, we’d like to conclude that *sec14l3* play important roles in zebrafish DA and PCV blood vessels formation via regulating angioblasts migration. We have also included these new results in our revised manuscript.

2. There are some issues with the analysis of the vascular phenotypes in the trunk of zebrafish embryos. The authors show that morpholino injected embryos show shortening of the DA, PCV and ISVs (Supplementary Figure 2). However, for the *sec14l3* mutants, they only analyze the length of the DA and PCV and not of the ISVs. In addition, they do not present the data as actual values (how many um), but only as a “relative” length. The authors need to provide the data for their length measurements. Is there a difference in ISV length between morpholino injected embryos and mutants? In addition, to verify the morpholino phenotype, the authors need to inject the morpholino into *sec14l3* mutants and measure ISV length. How do the values compare to the morpholino and mutant values?

Answer: We are sorry for the inconsistent data presentation with “relative” length values and have changed all of them to “actual” values in the revised manuscript (Fig. 2c, e, Fig. 3g, Fig. 4j, l and Supplementary Fig. 4). When analyzing ISVs formation, a typical process of angiogenesis, we found it was obviously affected in *sec14l3*

morphants but appeared almost normally in the mutant embryos. However, when *sec14l3*-tMO2 was injected in a *p53* mutant background, ISVs sprouting defect was largely restored (Fig. 2a and Supplementary Fig. 2), which presumably relieves the general off-target effect of morpholinos²⁶. What's more, according to the reviewer's suggestion, we injected *sec14l3*-tMO2 into embryos obtained from intercrossing *sec14l3* heterozygous and quantified DA, PCV and ISV lengths at 25 hpf and 30 hpf under different genotypes. After genotyping confirmation, we found that different from DA and PCV diameters without any more change, *sec14l3*-tMO2 could further reduce ISVs length in *sec14l3* homozygotes (Supplementary Fig. 4), suggesting that *sec14l3*-tMO2 indeed has a non-specific effect on ISVs sprouting but a specific effect on DA and PCV formation. We have included all of these results in our revised manuscript.

3. The authors also only show data for the lumen diameters in morpholino injected embryos, not in the mutants. The analysis for lumen diameters also needs to be repeated in mutants. From Figure 2d, it seems that ISV sprouting is only marginally affected (see point 2). Can the authors provide an explanation, why a process that heavily relies on VEGF signaling is only very mildly affected in their mutants? Can the authors analyze maternal/zygotic mutants, which might show a more severe phenotype in vascular development? In addition, a previous report showed that ERK phosphorylation affects endothelial migration in zebrafish ISVs (Shin et al., 2016, Development "Vegfa signals through ERK to promote angiogenesis, but not artery differentiation). The authors convincingly show reduction in ERK phosphorylation. Why is ISV migration not affected?

Answer: We appreciate the reviewer for these comments.

1) Actually, we have analyzed the DA and PCV defects of *sec14l3* mutant embryos with statistic data of lumen diameters in Figure 2d and 2e. Additionally, our new data double confirmed this conclusion (Supplementary Fig. 4).

2) As the reviewer noticed, there are only mildest ISV defects in *sec14l3* mutant embryos, which could not cause a significant difference in ISV length between the mutant and sibling embryos (Fig. 2d and Supplementary Fig. 4). Additionally, we found that the *MZsec14l3* mutants also show a similar phenotype with zygotic mutants in vascular development. It is well-known that VEGF signaling plays integral roles in both vasculogenesis and angiogenesis during vasculature. As for why the ISVs formation, a typical angiogenic process that heavily relies on VEGF signaling, is only very mildly affected in our mutants, it might be mainly due to the spatiotemporal expression pattern of *sec14l3*. As shown in Figure 1a, the whole mount in situ hybridization results revealed that *sec14l3* is expressed in both arterial and venous progenitors before the onset of the major axial vessels at 19 hpf, and thereafter restricted to the PCV but not DA. Because this particular time window is corresponding to vasculogenesis, but not angiogenesis context during vascular morphogenesis, we would like to prospect that *sec14l3* is not involved in the later process.

3) Nathan D. Lawson group reported that phosphorylation of Erk preferentially occurs in sprouting ISV endothelial cells, and both genetic and chemical inhibition of its phosphorylation prevent ISV endothelial sprouting²⁷. Due to the significant loss of pErk-positive cells in dorsal aorta, *kdrl* and *plcg1* mutant embryos both exhibit obvious defects in ISVs formation. However, in our case, knocking down or mutation of *sec14l3* couldn't lead to the loss of pErk-positive cells in dorsal aorta, but only a decrease of the p-Erk in these cells (Fig. 3d). We still could distinguish the clear pattern of p-Erk in sprouting ISV endothelial cells in *sec14l3* deficient embryos, which might be enough for the tip cells selection and subsequent ISVs formation. We have included these

discussions in our revised manuscript.

4. In HUVECs, both ERK and AKT phosphorylation are affected by *sec14l3* knockdown (Figure 3). For zebrafish, the authors only analyze ERK phosphorylation, but not AKT phosphorylation. Is AKT phosphorylation affected in mutant/morpholino injected zebrafish embryos?

Answer: We appreciate the reviewer for this comment and have performed immunostaining experiments using p-AKT antibody in *sec14l3* mutants. The results are shown in the Figure 3e. As expected, the p-AKT level, especially in the PCV, is obviously compromised in *sec14l3* deficient embryos. We have included these results in our revised manuscript.

5. The authors do not show any pictures depicting the vasculature of *ca-AKT* and *ca-MEK* injected embryos. How about overall body shape and blood flow in these embryos? Also, the description in the figure legend is not clear. Does the figure include the values from 3 independent experiments or just one experiment? Did the other two experiments show the same results? What values were used for statistical analysis? Are these from different fish in one experiment or different vessel regions of the same fish? Differential effects of *ca-AKT* and *ca-MEK* in different vessels is potentially an interesting finding. However, the authors do not address it further, neither experimentally, nor in the discussion.

Answer: We appreciate the reviewer for these comments.

1) In our revised manuscript, the pictures depicting the overall body shape and vasculature of *ca-AKT* and *ca-MEK* injected embryos have been shown along with the statistical data (Fig. 3f, g). After injection of 3 ng *sec14l3* tMO2, some non-specific cell apoptosis could be observed in the head region of the embryos without other obvious defects at 24 hpf. When *ca-AKT* or *ca-MEK* mRNA was co-injected with *sec14l3* tMO2 respectively or in combination, embryos display similarly with *sec14l3* morphants in morphology. In addition, it seems that only co-injection of *ca-MEK* and *ca-AKT* mRNA had a very slight rescue effect on the blood flow. According to the blood flow speed, the embryos were classified into three different phenotypes, group A with no blood circulation (normal blood production, data not shown), group B with weak circulation and group C with normal flow. At 30 hpf, knockdown of *sec14l3* would result in about 72.4% embryos in group A, 24.1% in group B and 3.5% in group C, while co-injection of 50 pg *ca-MEK* mRNA and 50 pg *ca-AKT* mRNA with *sec14l3* tMO2 led to 65.5%, 20.7% and 13.8% embryos in group A, B, C individually.

2) We feel very sorry about the unclear description of the figure legend. Actually, we performed these rescue experiments three times and only showed a set of representative data of once here. The two other repeats showed similar results. The DA and PCV values in statistical data here are from different fish in one experiment. To quantify the DA or PCV luminal diameter for an embryo, we used the average value of five independent measurements. We have rewritten this part in our revised manuscript.

3) About the distinct effects of ca-MEK and ca-AKT in different vessels, we need refer to the opposing roles of PLC γ /ERK and PI3K/AKT signaling in DA and PCV formation. It has been reported that within the arterial-fated angioblast, the VEGF ligand interacts with VEGFR2 to induce PLC γ activity and then stimulate the MEK/ERK kinase cascade, subsequently inducing expression of multiple genes in Notch pathway to ensure the acquisition of an arterial fate²⁸. Importantly, expression of constitutively active MEK or treatment with drugs that can activate PLC γ /ERK signaling can rescue DA formation in zebrafish embryos lacking *vegfa*, which is consistent with our results²⁹. Besides, Erk activation could also promote DA cell number and tight junction between arterial ECs³⁰. On the other hand, antagonism between PI3K/AKT and PLC γ /ERK signaling has been well documented³¹. In the venous-fated angioblast, PI3K/AKT signaling is activated to antagonize ERK activity, and overexpression of constitutive active Akt promotes the venous cell fate. Therefore, in *sec14l3* mutant embryos bearing a relatively low level of VEGF activity, overexpression of *ca-MEK* or *ca-AKT* could rescue the DA or PCV formation respectively, which just seems to make sense. Since both PLC γ /ERK and PI3K/AKT pathways are activated downstream of VEGF receptor, how they are preferentially activated in different endothelial progenitor cells remain intriguing questions. Furthermore, how *Sec14l3* exactly regulates these processes need to be investigated in the future. We thank the reviewer very much for this constructive suggestions and have added these discussions in our revised manuscript.

6. The authors claim that *sec14l3* limits the exposure of phosphorylated VEGFR2 to phosphatases, such as PTP1B. To distinguish between phosphatase exposure and a possible role of *sec14l3* in protein degradation, the authors need to analyze the effect of inhibiting protein degradation on VEGFR2 phosphorylation after knockdown of *sec14l3* (e.g. via inhibiting the proteasome).

Answer: We thank the reviewer for this constructive advice. Actually, we found that *SEC14L2* knockdown not only compromised the internalization and recycling of VEGFR2, but also led to the accumulation of VEGFR2 in CD63 or

RAB7 expressing late endosomes in HUVECs (Fig. 5e, f). However, our western blotting results didn't show any effect of SEC14L2 knockdown on the VEGFR2 protein level (Supplementary Fig. 11). Combining these two results together, we would like to believe that more VEGFR2 is stored in late endosomes but not delivered to the lysosome or proteasome for degradation in SEC14L2 deficient cells, which is consistent with previous reports¹¹.

To further exclude the possibility that SEC14L2 modulates VEGFR2 activity by regulating its degradation, we detected the p-Y1175 and total VEGFR2 protein levels in SEC14L2 knockdown cells after incubation of proteasome inhibitor MG132. Results showed that SEC14L2 knockdown could also lead to the reduction of p-VEGFR2 at Y1175 site when using MG132 to block protein degradation, further supporting the conclusion that SEC14L2 limits the exposure of p-VEGFR2 to phosphatases, but not prevents its degradation.

7. The authors use a complex biotin labelling assay to analyse VEGFR2 recycling (Fig. 5d). The lysates IN2 show a reduced amount of biotinylated VEGFR2 in NC-shRNA infected cells compared to IN1 lysate. This observation is attributed to VEGFR2 recycling to the cell membrane. However, can the authors exclude the degradation of VEGFR2 upon internalization since they write in the discussion that VEGFR2 can undergo lysosomal degradation? Is it possible to observe the same changes in intracellular VEGFR2 upon inhibition of protein degradation? Also, is it possible to measure biotinylated VEGFR2 only on the plasma membrane (before and after internalization)? The interpretation of the assay is important for understanding the role of SEC14L2, which shows a clear effect in this experiment. Could you also explain why there seems to be more biotinylated VEGFR2 upon internalization (IN1) than before (PM).

Answer: We thank the reviewer for these comments.

1) In the recycling assay, we induced internalization at 22°C, but not 37°C, which could allow the internalized VEGFR2 deliver to the RAB4⁺ compartment for recycling, but not late endosome or lysosome for degradation¹¹. That's to say, this treatment excludes the degradation effect of VEGFR2. Therefore, this method could be used to quantify the recycling rate of VEGFR2.

2) About the measurement of biotinylated PM VEGFR2 after internalization, it is impossible for us to do that since there is no appropriate method to specifically deplete the internalized intracellular biotinylated VEGFR2, which could interface the quantification of biotinylated VEGFR2 left at the PM after internalization. Conversely, PM-biotin proteins could be removed by membrane-impermeant reducing agent without effect on internalized proteins, then it is easy for us to quantify the internalized intracellular biotinylated VEGFR2 in our assay.

3) About the Figure 6d, we feel very sorry for this mistake, we have double checked the result and revised the data in our new version.

8. The authors claim that sec14l3 preferentially binds to GDP bound RAB4A and RAB5A performing binding assays in the presence of GTPgammaS. However, in a previous report (Gong et al., 2017), the authors showed that sec14l3 itself shows GTPase activity. How can they exclude that the GTPgammaS inhibits sec14l3 GTPase activity

and thereby diminishes interaction with RAB4A and RAB5A?

Answer: GTPgammaS is an analog of GTP, which should act as an agonist but not an antagonist of the Sec14I3 GTPase activity. Therefore, we could exclude this possibility.

Minor:

1. The authors motivate their study with an RNA-Seq experiment, however they do not show any results. It is also not clear, whether the expression of sec14I3 was detected using RNA-Seq. The results should be either shown or the RNA-Seq should not be mentioned.

Answer: We thank the reviewer for this suggestion and have included the RNA-seq data in our revised manuscript as Supplementary data 1.

2. In the abstract the authors state “However, little is known about how Sec14-like phosphatidylinositol transfer proteins (PITPs) are involved in this process.” It is not obvious why PITPs would be involved in VEGF signalling. An additional sentence about the study rationale could make it clearer.

Answer: We thank the reviewer for this advice. What we want to express here is whether Sec14-like phosphatidylinositol transfer proteins are involved in VEGF signaling. We have rewritten this sentence in the revised manuscript.

3. The description of the biotinylation assay should be moved from the results section to the materials and methods section.

Answer: We thank the reviewer for this advice and have moved the responding part to the materials and methods section.

4. The sentence in the Discussion “Of these known regulators, either VEGFR2 internalization or degradation is affected” could be rephrased to improve the clarity.

Answer: We thank the reviewer for this comment and have rewritten this sentence in the new version.

References

1. Lampugnani MG, Orsenigo F, Gagliani MC, Tacchetti C, Dejana E. Vascular endothelial cadherin controls VEGFR-2 internalization and signaling from intracellular compartments. *J Cell Biol* **174**, 593-604 (2006).
2. Mattila E, Auvinen K, Salmi M, Ivaska J. The protein tyrosine phosphatase TCPTP controls VEGFR2 signalling. *J Cell Sci* **121**, 3570-3580 (2008).
3. Mellberg S, *et al.* Transcriptional profiling reveals a critical role for tyrosine phosphatase VE-PTP in regulation of VEGFR2 activity and endothelial cell morphogenesis. *FASEB J* **23**, 1490-1502 (2009).
4. Mitola S, *et al.* Type I collagen limits VEGFR-2 signaling by a SHP2 protein-tyrosine phosphatase-dependent mechanism 1. *Circ Res* **98**, 45-54 (2006).
5. Nakamura Y, *et al.* Role of protein tyrosine phosphatase 1B in vascular endothelial growth factor signaling and cell-cell adhesions in endothelial cells. *Circ Res* **102**, 1182-1191 (2008).

6. Nottebaum AF, *et al.* VE-PTP maintains the endothelial barrier via plakoglobin and becomes dissociated from VE-cadherin by leukocytes and by VEGF. *J Exp Med* **205**, 2929-2945 (2008).
7. Presta M, Dell'Era P, Mitola S, Moroni E, Ronca R, Rusnati M. Fibroblast growth factor/fibroblast growth factor receptor system in angiogenesis. *Cytokine Growth Factor Rev* **16**, 159-178 (2005).
8. Murakami M, *et al.* FGF-dependent regulation of VEGF receptor 2 expression in mice. *J Clin Invest* **121**, 2668-2678 (2011).
9. Murakami M, *et al.* The FGF system has a key role in regulating vascular integrity. *Journal of Clinical Investigation* **118**, 3355-3366 (2008).
10. dela Paz NG, Walshe TE, Leach LL, Saint-Geniez M, D'Amore PA. Role of shear-stress-induced VEGF expression in endothelial cell survival. *J Cell Sci* **125**, 831-843 (2012).
11. Gampel A, Moss L, Jones MC, Brunton V, Norman JC, Mellor H. VEGF regulates the mobilization of VEGFR2/KDR from an intracellular endothelial storage compartment. *Blood* **108**, 2624-2631 (2006).
12. Manickam V, *et al.* Regulation of vascular endothelial growth factor receptor 2 trafficking and angiogenesis by Golgi localized t-SNARE syntaxin 6. *Blood* **117**, 1425-1435 (2011).
13. Eberth A, Ahmadian MR. In vitro GEF and GAP assays. *Curr Protoc Cell Biol* **Chapter 14**, Unit 14 19 (2009).
14. Kanie T, Jackson PK. Guanine Nucleotide Exchange Assay Using Fluorescent MANT-GDP. *Bio Protoc* **8**, (2018).
15. Shin D, *et al.* Site-specific monoubiquitination downregulates Rab5 by disrupting effector binding and guanine nucleotide conversion. *Elife* **6**, (2017).
16. Yeh BJ, Rutigliano RJ, Deb A, Bar-Sagi D, Lim WA. Rewiring cellular morphology pathways with synthetic guanine nucleotide exchange factors. *Nature* **447**, 596-600 (2007).
17. Gong B, Shen W, Xiao W, Meng Y, Meng A, Jia S. The Sec14-like phosphatidylinositol transfer proteins Sec14I3/SEC14L2 act as GTPase proteins to mediate Wnt/Ca(2+) signaling. *Elife* **6**, (2017).
18. Barr F, Lambright DG. Rab GEFs and GAPs. *Curr Opin Cell Biol* **22**, 461-470 (2010).
19. Blumer J, *et al.* RabGEFs are a major determinant for specific Rab membrane targeting. *J Cell Biol* **200**, 287-300 (2013).
20. Horiuchi H, *et al.* A novel Rab5 GDP/GTP exchange factor complexed to Rabaptin-5 links nucleotide exchange to effector recruitment and function. *Cell* **90**, 1149-1159 (1997).
21. Muller MP, Goody RS. Molecular control of Rab activity by GEFs, GAPs and GDI. *Small GTPases*, 1-17 (2017).
22. Nottingham RM, Pfeffer SR. Defining the boundaries: Rab GEFs and GAPs. *Proc Natl Acad Sci U S A* **106**, 14185-14186 (2009).
23. Zhang Z, *et al.* Molecular mechanism for Rabex-5 GEF activation by Rabaptin-5. *Elife* **3**, (2014).
24. Kohli V, Schumacher JA, Desai SP, Rehn K, Sumanas S. Arterial and venous progenitors of the major axial vessels originate at distinct locations. *Dev Cell* **25**, 196-206 (2013).

25. Herbert SP, *et al.* Arterial-venous segregation by selective cell sprouting: an alternative mode of blood vessel formation. *Science* **326**, 294-298 (2009).
26. Robu ME, *et al.* p53 activation by knockdown technologies. *PLoS Genet* **3**, e78 (2007).
27. Shin M, Beane TJ, Quillien A, Male I, Zhu LJ, Lawson ND. Vegfa signals through ERK to promote angiogenesis, but not artery differentiation. *Development* **143**, 3796-3805 (2016).
28. Rossant J. Vascular development and patterning: making the right choices. *Current Opinion in Genetics & Development* **13**, 408-412 (2003).
29. Kim SH, Schmitt CE, Woolls MJ, Holland MB, Kim JD, Jin SW. Vascular endothelial growth factor signaling regulates the segregation of artery and vein via ERK activity during vascular development. *Biochem Biophys Res Commun* **430**, 1212-1216 (2013).
30. Zhang C, *et al.* Inhibition of endothelial ERK signalling by Smad1/5 is essential for haematopoietic stem cell emergence. *Nat Commun* **5**, 3431 (2014).
31. Hong CC, Peterson QP, Hong JY, Peterson RT. Artery/vein specification is governed by opposing phosphatidylinositol-3 kinase and MAP kinase/ERK signaling. *Curr Biol* **16**, 1366-1372 (2006).

REVIEWERS' COMMENTS:

Reviewer #1 (Remarks to the Author):

In the original manuscript, I raised 4 potential problems of their manuscript for publication by Nature Communications.

- 1) Lack of a great conceptual advancement
- 2) Specificity of the upstream signaling, especially in the context of shear stress.
- 3) Specific effect on VEGFR2 Y1175 but not on Y1054/1059.
- 4) In vitro experimental design using HUVEC.

In the revised manuscript, the authors addressed 3rd point very in a convincing way. Additionally, the manuscript was improved after addressing other reviewers' comments. However, other points I raised were not seriously addressed.

First point was unfortunately not addressed at all.

Rescue experiments by VEGFR2-cyto mRNA is great. But I feel this is not enough to address my concern. In page 12, author's described the role of VEGFR2-induced PLCgamma/ERK kinase signal cascade on DA formation. Recently, same authors published the manuscript focusing on Sec14I3/SEC14L2 on PLCgamma activation downstream of WNT/Ca²⁺ signaling in zebrafish (Elife 2017), suggesting Sec14I3/SEC14L2 has many signaling upstream.

Particularly in the context of shear stress would be important, as shear stress on arterial ECs are greater than that on venous ECs. Many previous works addressed the role of shear stress on VEGFR signaling. Shear stress induces VEGFR2 activation. So, are in vitro experiments mainly addressed VEGF-induced VEGFR2 activation proper experiments to address the phenotype observed in vivo?

My 4th criticism is related to these comments.

Therefore, I could not support this manuscript for publication by Nature Communications.

Minor comment

All of the data must be shown according to the policy of Nature Communications. Page 8, line 203.

Reviewer #2 (Remarks to the Author):

The authors have responded to the question of how SEC14L2/Sec14I3 potentiates RAB5A/4A activity by excluding the obvious possibility that SEC14L2/Sec14I3 is not acting as a GTP exchange factor (GEF) (new results provided in Supplementary Fig. 12). Unfortunately, the authors do not provide an insight into how SEC14L2/Sec14I3 potentiates RAB5A/4A activity. This is a weakness considering that this stated in the Title.

They state that "Mechanistically, sec14I3 promotes RAB5A/4A activation" (line 73 and elsewhere). Activation of RABs imply that the GTP-bound state is increased but then in Line 17, in the abstract they claim that Sec14L2 directly binds to RAB5A/4A and promotes their GTPase activity. (See also line 343 where again they state that RAB5A/RAB4A modulate their GTPase activity). Surely they mean that it promotes an increase in the GTP-bound state of RABs, not the GTPase activity as this would lead to a decrease from the GTP state to a GDP state. These two statements are contradictory and needs clarification.

The discussion on how Sec14I3/SEC14L2 (lines 347-356) seems rather superfluous and does not really provide any insights into the mechanism of RAB4/RAB5 activation.

Reviewer #4 (Remarks to the Author):

The revised version is significantly improved and I only have some minor comments.

1. Page 7: PTP1b is a specific "scrambler". It is not clear to me, what "scrambler" would signify in this context.
2. Figure 4f: Why is there a band in the second lane (anti-HA), even though these cells were not transfected with an HA construct?
3. The authors motivate Figure 6a to test to which VEGFR Sec14I3 binds. However, in Figure 4f they already use the VEGFR2 cyto construct. Logically, Figure 6a should be before Figure 4f.
4. References for ca-ERK, ca-MEK and VEGFR2 kinase dead mutant constructs seem to be missing.
5. Supplementary Fig. 2 still only contains "relative lengths" measurements for ISVs instead of actual lengths measurements.

Reviewer #1 (Remarks to the Author):

In the original manuscript, I raised 4 potential problems of their manuscript for publication by Nature Communications.

- 1) Lack of a great conceptual advancement
- 2) Specificity of the upstream signaling, especially in the context of shear stress.
- 3) Specific effect on VEGFR2 Y1175 but not on Y1054/1059.
- 4) In vitro experimental design using HUVEC.

In the revised manuscript, the authors addressed 3rd point very in a convincing way. Additionally, the manuscript was improved after addressing other reviewers' comments. However, other points I raised were not seriously addressed. First point was unfortunately not addressed at all.

Answer: We are sorry for not fully discussing the conceptual advancement of our study in the previous response letter. Our key findings can be summarized into two aspects as follows.

1) The essential roles of Sec14I3 in embryonic vascular development. Phosphatidylinositol transfer proteins (PITPs) are implicated in the traffic of phosphoinositides (PIs) between different membrane compartments, but to what biological outcomes are coupled remains rarely reported. By establishing a CRISPR/Cas9-generated zebrafish mutant *sec14I3*, a family member of class III PITPs, we explored its physiological functions in regulating DA and PCV formation during embryonic vasculogenesis. So far two distinct mechanisms have been proposed to account for the DA and PCV formation in zebrafish. In 2013, Kohli V et al., demonstrated that arterial and venous progenitors of the major axial vessels originate at distinct locations, while Herbert SP et al., reported an alternative mode that arterial-venous segregation by selective cell sprouting from the vascular cord in 2009^{1,2}. However, we found that both the migration of angioblasts at 14-16 somite stages and the sprouting of venous progenitors from the vascular cord at 21-23 hpf are decreased obviously in *sec14I3* depleted embryos (Fig. 2f, g, and Supplementary Fig. 6). Therefore, to some extent, we would like to suggest that these two models exist simultaneously in a unified manner.

2) The critical functions and possible mechanisms of Sec14I3 in regulating VEGFR2 endocytic trafficking. Our work here for the first time provides the missing link between Sec14I3 and VEGFR2 in a physiological context. Even though several factors have been reported to regulate VEGFR2 trafficking, it is not known whether and how phosphatidylinositol transfer proteins (PITPs) can participate in this process. Here, we show that Sec14I3/SEC14L2 physically bind to VEGFR2 and promote its internalization and recycling, therefore potentiating its activation through protecting p-VEGFR2-Y¹¹⁷⁵ from dephosphorylation by the peri-membrane tyrosine phosphatase PTP1B. Meanwhile, Sec14I3/SEC14L2 interact with RAB5A/4A and facilitate their GTP-bound states formation, which might be required for regulating VEGFR2 endocytic trafficking. Furthermore, due to the

involvement of PI molecules in endocytic membrane trafficking, whether PIPs integrate lipid metabolism with intracellular VEGFR2 signaling will be another interesting question to be explored in the future.

We have revised the abstract and discussion parts to address the conceptual advancement of our study in the revised manuscript.

Rescue experiments by VEGFR2-cyto mRNA is great. But I feel this is not enough to address my concern. In page 12, author's described the role of VEGFR2-induced PLCgamma/ERK kinase signal cascade on DA formation. Recently, same authors published the manuscript focusing on Sec14I3/SEC14L2 on PLCgamma activation downstream of WNT/Ca2+ signaling in zebrafish (Elife 2017), suggesting Sec14I3/SEC14L2 has many signaling upstream.

Particularly in the context of shear stress would be important, as shear stress on arterial ECs are greater than that on venous ECs. Many previous works addressed the role of shear stress on VEGFR signaling. Shear stress induces VEGFR2 activation. So, are *in vitro* experiments mainly addressed VEGF-induced VEGFR2 activation proper experiments to address the phenotype observed *in vivo*? My 4th criticism is related to these comments.

Answer: We fully understand the reviewer's concern and feel sorry about not interpreting it clearly in our previous version. As the reviewer mentioned, besides VEGF ligands, PECAM-1 mediated fluid shear stress (FSS) signaling could also recruit VEGFR2 and then phosphorylate it at Y¹¹⁷⁵ site along with VE-Cadherin, thus regulating vascular remodeling, vascular homeostasis, cardiac development and atherogenesis³. In this study, we didn't focus on FSS-induced VEGFR2 signaling, largely due to our *in vivo* observations that *sec14I3* specifically exists and exerts its function on angioblast migration during zebrafish vasculogenesis before the onset of blood flow (Fig. 1a, b, 2f, g, and Supplementary Fig. 6). Additionally, different from mammalian vascular development, it has been reported that changes in shear stress appear to be relatively unimportant in the initial stages of zebrafish angiogenesis because blood vessels can be successfully formed in the absence of heartbeat and blood flow until 14 days postfertilization⁴. Therefore, based on the essential roles of VEGF ligands in angioblast migration during vasculogenesis^{5,6,7}, we chose VEGFa as a stimulus to activate VEGFR2 signaling transduction *in vitro*, and further investigated *SEC14L2* knockdown effect in HUVECs and HUAECs on the downstream effectors of VEGF signaling to mimic the endogenous regulation. Even so, we still cannot exclude that Sec14I3 could participate in regulating FSS-induced VEGFR2 activation during later stages of zebrafish angiogenesis. It will be interesting to explore this possibility in the future. We have rewritten the corresponding contents of the result and discussion parts in this version.

Minor comment

All of the data must be shown according to the policy of Nature Communications. Page 8, line 203.

Answer: Thanks for the reviewer's reminding and we have added the data in Supplementary Fig. 8 in the new version.

Reviewer #2 (Remarks to the Author):

The authors have responded to the question of how SEC14L2/Sec14I3 potentiates RAB5A/4A activity by excluding the obvious possibility that SEC14L2/Sec14I3 is not acting as a GTP exchange factor (GEF) (new results provided in Supplementary Fig. 12). Unfortunately, the authors do not provide an insight into how SEC14L2/Sec14I3 potentiates RAB5A/4A activity. This is a weakness considering that this is stated in the Title. They state that "Mechanistically, sec14I3 promotes RAB5A/4A activation" (line 73 and elsewhere). Activation of RABs imply that the GTP-bound state is increased but then in Line 17, in the abstract they claim that Sec14L2 directly binds to RAB5A/4A and promotes their GTPase activity. (See also line 343 where again they state that RAB5A/RAB4A modulate their GTPase activity). Surely they mean that it promotes an increase in the GTP-bound state of RABs, not the GTPase activity as this would lead to a decrease from the GTP state to a GDP state. These two statements are contradictory and need clarification. The discussion on how Sec14I3/SEC14L2 (lines 347-356) seems rather superfluous and does not really provide any insights into the mechanism of RAB4/RAB5 activation.

Answer: We appreciate the reviewer for this valuable comment. As the reviewer mentioned, when referring to the GTPase activity of a G protein, it always should take two kinetic characteristics into consideration independently and separately. One is its GTP uptake activity and the other is GTP hydrolysis activity, which can be measured respectively^{8,9}. In this study, it was suggested that Sec14I3 could directly interact with RAB5A/4A and obviously promote their GTP-bound state formation (Fig. 7c, d), although Sec14I3 could not act as a GEF for RAB5 directly proved by the *in vitro* exchange assay based on mant-GTPγS fluorescence (Supplementary Fig. 12). Actually, to determine whether Sec14I3 could regulate the GTP hydrolysis activity of RAB5A/4A, MESG-based single-turnover assay needs to be performed further. We are sorry for not describing these two independent processes clearly and sometimes summarizing it as GTPase activity generally. We have already revised these statements in the new version.

Based on our data here, we would like to propose the following potential mechanisms of SEC14L2/Sec14I3 acting on RAB5A/4A activation. 1) Sec14I3 might serve as an adaptor to recruit a GEF protein for RAB5A/4A. Although we haven't detected any interaction between Sec14I3 and several well-known GEF proteins of RAB5, such as Rabex5 and RABGEF1, we still cannot exclude this possibility, since there might be other unidentified GEFs for RAB5A/4A that could participate in this process. 2) Sec14I3 belongs to atypical class III phosphatidylinositol transfer proteins (PITPs), which are implicated in the traffic of phosphoinositides (PIs) between different membrane compartments¹⁰. Different PI species display distinct subcellular distributions where they play intriguing functions. For example, PI(3)P, PI(4)P and PI(3,5)P₂ are found predominantly within

early, recycling and late endosomes respectively¹¹, which are essential for appropriate endosomal trafficking events. *Drosophila* PI 3-phosphatase MTMR13 promotes PI(3)P turnover at endosomes to activate Rab5 family member Rab21, PI(3)P levels in early endosome may drive RAB5-RAB7 conversion and so on.^{12,13}. Therefore, it deserves further exploration of whether Sec14I3 could also motivate specific phosphoinositides transportation along vesicles to regulate endosome dynamics, either in morphology or in effector recruitment. Additionally, whether the intrinsic GTPase activity of Sec14I3 is required for the activation of RAB5A/4A also awaits further investigation. We have improved the corresponding discussion part in our revised manuscript.

Reviewer #4 (Remarks to the Author):

The revised version is significantly improved and I only have some minor comments.

1. Page 7: PTP1b is a specific “scrambler”. It is not clear to me, what “scrambler” would signify in this context.

Answer: We thank the reviewer very much for this criticism and feel very sorry for this mistake. Actually, we would have liked to use “scavenger” to signify that PTP1b functions as a phosphatase to remove the phosphorylation state of VEGFR2 at Y¹¹⁷⁵ site. To be more precise, we rewrote this sentence to “PTP1B has been reported and identified as such a specific phosphatase” in the revised manuscript.

2. Figure 4f: Why is there a band in the second lane (anti-HA), even though these cells were not transfected with an HA construct?

Answer: We appreciate the reviewer for this comment. This band is corresponding to the IgG heavy chain from the HA antibody. Actually, the other four lanes also have this band, but they are all covered by the HA-PTP1B band which is also around 55 kDa and thus very close to the IgG heavy chain.

3. The authors motivate Figure 6a to test to which VEGFR Sec14I3 binds. However, in Figure 4f they already use the VEGFR2 cyto construct. Logically, Figure 6a should be before Figure 4f.

Answer: We thank the reviewer for this comment. As the reviewer suggested, we have moved Figure 6a before Figure 4f, and then split the affected Figure 4 to make the writing more logically.

4. References for ca-ERK, ca-MEK and VEGFR2 kinase dead mutant constructs seem to be missing.

Answer: Thanks for the reviewer’s reminding and we have added the related information and references in the revised version.

5. Supplementary Fig. 2 still only contains “relative lengths” measurements for ISVs instead of actual lengths measurements.

Answer: We are sorry for the inconsistent data presentation with “relative lengths”, and have changed them to “actual lengths” in the revised manuscript (Supplementary Fig. 2).

References:

1. Kohli V, Schumacher JA, Desai SP, Rehn K, Sumanas S. Arterial and venous progenitors of the major axial vessels originate at distinct locations. *Dev Cell* **25**, 196-206 (2013).
2. Herbert SP, *et al.* Arterial-venous segregation by selective cell sprouting: an alternative mode of blood vessel formation. *Science* **326**, 294-298 (2009).
3. Tzima E, *et al.* A mechanosensory complex that mediates the endothelial cell response to fluid shear stress. *Nature* **437**, 426-431 (2005).
4. Pelster B, Burggren WW. Disruption of hemoglobin oxygen transport does not impact oxygen-dependent physiological processes in developing embryos of zebra fish (*Danio rerio*). *Circ Res* **79**, 358-362 (1996).
5. Liang D, *et al.* Cloning and characterization of vascular endothelial growth factor (VEGF) from zebrafish, *Danio rerio*. *Biochim Biophys Acta* **1397**, 14-20 (1998).
6. Nasevicius A, Larson J, Ekker SC. Distinct requirements for zebrafish angiogenesis revealed by a VEGF-A morphant. *Yeast* **17**, 294-301 (2000).
7. Liang D, *et al.* The role of vascular endothelial growth factor (VEGF) in vasculogenesis, angiogenesis, and hematopoiesis in zebrafish development. *Mech Dev* **108**, 29-43 (2001).
8. Vetter IR, Wittinghofer A. The guanine nucleotide-binding switch in three dimensions. *Science* **294**, 1299-1304 (2001).
9. Gong B, Shen W, Xiao W, Meng Y, Meng A, Jia S. The Sec14-like phosphatidylinositol transfer proteins Sec14I3/SEC14L2 act as GTPase proteins to mediate Wnt/Ca(2+) signaling. *Elife* **6**, (2017).
10. Wiedemann C, Cockcroft S. The Role of Phosphatidylinositol Transfer Proteins (PITPs) in Intracellular Signalling. *Trends Endocrinol Metab* **9**, 324-328 (1998).
11. De Matteis MA, Godi A. PI-loting membrane traffic. *Nat Cell Biol* **6**, 487-492 (2004).
12. Rink J, Ghigo E, Kalaidzidis Y, Zerial M. Rab conversion as a mechanism of progression from early to late endosomes. *Cell* **122**, 735-749 (2005).
13. Jean S, Cox S, Schmidt EJ, Robinson FL, Kiger A. Sbf/MTMR13 coordinates PI(3)P and Rab21 regulation in endocytic control of cellular remodeling. *Mol Biol Cell* **23**, 2723-2740 (2012).